# LRG1 is an adipokine that promotes insulin sensitivity and suppresses inflammation

Chan Hee J Choi[1,2], William Barr[2†], Samir Zaman[2†], Corey Model[2], Annsea Park[3], Mascha Koenen[2], Zeran Lin[2], Sarah K Szwed[1,2], Francois Marchildon[2], Audrey Crane[2], Thomas S Carroll[4], Henrik Molina[5], Paul Cohen[2*]

[1]Weill Cornell/Rockefeller/Sloan Kettering Tri-Institutional MD-PhD Program, New York, United States; [2]Laboratory of Molecular Metabolism, Rockefeller University, New York, United States; [3]Department of Immunobiology, Yale University, New Haven, United States; [4]Bioinformatics Resouce Center, Rockefeller University, New York, United States; [5]Proteomics Resource Center, Rockefeller University, New York, United States

**\*For correspondence:**
pcohen@rockefeller.edu

†These authors contributed equally to this work

**Competing interest:** The authors declare that no competing interests exist.

**Abstract** While dysregulation of adipocyte endocrine function plays a central role in obesity and its complications, the vast majority of adipokines remain uncharacterized. We employed bio-orthogonal non-canonical amino acid tagging (BONCAT) and mass spectrometry to comprehensively characterize the secretome of murine visceral and subcutaneous white and interscapular brown adipocytes. Over 600 proteins were identified, the majority of which showed cell type-specific enrichment. We here describe a metabolic role for leucine-rich α–2 glycoprotein 1 (LRG1) as an obesity-regulated adipokine secreted by mature adipocytes. LRG1 overexpression significantly improved glucose homeostasis in diet-induced and genetically obese mice. This was associated with markedly reduced white adipose tissue macrophage accumulation and systemic inflammation. Mechanistically, we found LRG1 binds cytochrome c in circulation to dampen its pro-inflammatory effect. These data support a new role for LRG1 as an insulin sensitizer with therapeutic potential given its immunomodulatory function at the nexus of obesity, inflammation, and associated pathology.

## Editor's evaluation

This paper presents a fundamental advance by elucidating the function of LRG1 as an adipokine. The authors provide compelling evidence for the biological effects of LRG1 on metabolism and its potential connections to metabolic diseases.

## Introduction

Obesity is a major threat to human health due to its association with serious comorbidities (*Angelantonio et al., 2016*; *Poirier et al., 2006*). Now considered an important endocrine organ, adipose tissue secretes a constellation of bioactive peptides, or adipokines, many of which regulate whole-body energy homeostasis and inflammation (*Funcke and Scherer, 2019*). Dysregulation of adipose tissue endocrine function is a key feature of obesity and a major contributor to its sequelae. However, the vast majority of adipokines remain unstudied. Addressing this knowledge gap calls for a detailed characterization of the adipose secretome to better understand how adipocytes communicate with other cells to coordinate systemic metabolism.

Mammals possess white (WAT) and brown adipose tissues (BAT), with divergent effects on whole-body metabolism. Visceral (Visc) WAT is particularly associated with obesity-related diseases, whereas subcutaneous (SubQ) WAT is comparatively benign (*Fox et al., 2007*). On the other hand, active BAT is associated with improved cardiometabolic health (*Becher et al., 2021*; *Lee et al., 2014*). While Visc fat contains predominantly white adipocytes that efficiently store energy, SubQ depots contain a mixture of white and thermogenic beige adipocytes which, like brown adipocytes, dissipate energy as heat via uncoupled respiration (*Chouchani et al., 2019*). In addition to these bioenergetic properties, transplantation studies in mice have demonstrated that secretory mediators likely convey the metabolic effects of different adipose tissues (*Tran et al., 2008*). Moreover, BAT-specific secreted factors, called batokines, are increasingly appreciated as important contributors to the benefits associated with BAT activation (*Villarroya et al., 2017*).

Transcriptomic profiling coupled with secretion prediction algorithms is often utilized to identify putative secreted proteins, but this approach could omit non-classically secreted factors. Alternatively, direct detection of proteins in conditioned medium (CM) can be performed using mass spectrometry (MS). Such proteomic analyses are commonly performed in fetal bovine serum (FBS)-free conditions because FBS proteins interfere with detection of relatively low abundance secreted proteins in CM. Serum starvation, however, induces a complex and unpredictable response from cultured cells, making physiological interpretation difficult (*Pirkmajer and Chibalin, 2011*).

To overcome these limitations, a chemoproteomic technology called BONCAT has been applied for secretome profiling (*Dieterich et al., 2006*; *Eichelbaum et al., 2012*). L-Azidohomoalanine (AHA) is a non-toxic, non-canonical amino acid with structural similarity to L-methionine (Met) (*Figure 1A*). AHA can be recognized by cells' native translational machinery and incorporated into the nascent proteome in place of Met. The azide functional group in AHA does not exist in nature, so peptides containing AHA can be targeted for bio-orthogonal chemical conjugation strategies called click chemistry. Applying this technique to cultured adipocytes allows for selective enrichment of low abundance nascent secreted proteins for proteomic analysis of serum-containing CM.

Here, we employed BONCAT to comprehensively profile the adipose secretome and nascent serum proteome of mice, from which we identified a novel adipokine, leucine-rich α–2 glycoprotein 1 (LRG1). LRG1 is secreted by mature adipocytes and increased in obesity. We demonstrate that LRG1 improves fasting glucose and insulin tolerance and reduces adipose tissue macrophage accumulation. At the molecular level, LRG1 binds extracellular cytochrome *c* (Cyt *c*) released from dead/dying cells and dampens Cyt *c*'s pro-inflammatory effect on macrophages. These data reveal LRG1 as a crucial regulator of metabolic health as an insulin sensitizer and suppressor of systemic inflammation in obesity.

## Results

### Characterization of primary adipocyte secretome

To profile the secretome of different types of primary adipocytes, stromal vascular fraction (SVF) from epididymal white adipose tissue (eWAT), inguinal white adipose tissue (iWAT), and interscapular brown adipose tissue (BAT) from C57BL/6 J (B6) mice was differentiated in vitro into primary Visc, SubQ, and Brown adipocytes, respectively (*Figure 1B*). By day 6 of differentiation, all three cell types showed comparable lipid droplet accumulation and expression of mature adipocyte markers such as *Fabp4*, *Pparg2*, and *Adipoq* (*Figure 1—figure supplement 1A, B*). In vitro primary adipocytes recapitulated characteristic gene expression patterns in vivo, with expression of *Prdm16* and *Ucp1* highest in Brown, intermediate in SubQ, and lowest in Visc (*Figure 1—figure supplement 1C*). Adipocytes were pulsed with Met-deficient media containing 0.1 mM AHA and 10% FBS for 24 hours, after which CM was collected. Except for *Pparg2*, we did not observe a significant effect of AHA pulse on expression of mature adipocyte or thermogenic genes (*Figure 1—figure supplement 1D*). To visualize azide-labeled secreted proteins, CM was subjected to a copper-catalyzed azide-alkyne cycloaddition (CuAAC) reaction with tetramethylrhodamine (TAMRA)-alkyne. In-gel fluorescence analysis highlighted adipocyte-derived proteins in CM as bands positive for TAMRA fluorescence (*Figure 1C*).

For identification and quantitative analysis of the secretome across the three adipose cell types, azide-labeled proteins were enriched from CM by conjugating with alkyne-agarose beads. On-bead digested peptides were subjected to label-free MS. We detected a total of 742 proteins, of which 138 were excluded as reverse hits or contaminants (*Figure 1—figure supplement 1E*). The intensity-based

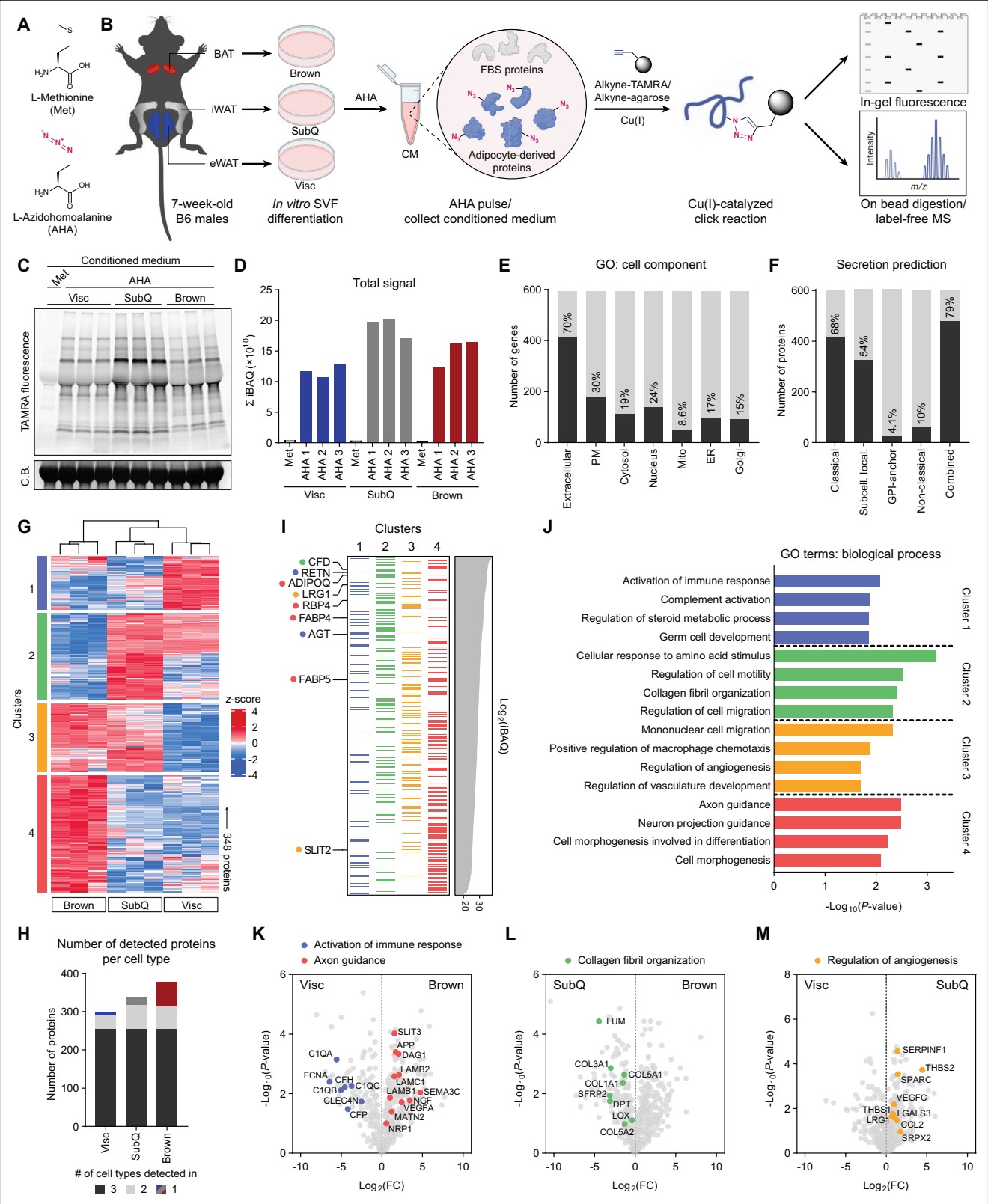

**Figure 1.** Characterization of primary adipocyte secretome using BONCAT. (**A**) Chemical structures of L-methionine (Met) and L-azidohomoalanine (AHA). (**B**) Schematic diagram of MS-based secretome analysis from primary Visc, SubQ, and Brown adipocytes using BONCAT. (**C**) In-gel fluorescence analysis of TAMRA-conjugated CM proteins from Met-pulsed (lane 1) or AHA-pulsed (lanes 2–10) adipocytes. C.B., Coomassie Blue. (**D**) Sum of iBAQ intensities ($\Sigma$ iBAQ) across all quantified proteins in each CM. (**E**) Number and proportion of genes annotated to GO cell component terms.

*Figure 1 continued on next page*

*Figure 1 continued*

Extracellular, extracellular region, space, or matrix; PM, plasma membrane; Mito, mitochondrion; ER, endoplasmic reticulum; Golgi, Golgi apparatus. (**F**) Number and proportion of proteins predicted to be secreted by prediction algorithms. (**G**) Heatmap of 348 differentially secreted proteins across cell types. (**H**) Number of proteins detected in at least two of three biological replicates per cell type, color-coded based on number of cell types a protein is detected in. (**I**) 348 proteins grouped in clusters from (**G**) and arranged in decreasing order of iBAQ intensities. Previously described adipokines are indicated. (**J**) Top 4 overrepresented GO biological process terms per cluster and their enrichment scores. (**K–M**) Pairwise comparisons of Log₂(LFQ) intensities between Visc and Brown (**K**), SubQ and Brown (**L**), and Visc and SubQ (**M**) CM. Proteins and their annotated pathway terms from (**J**) are indicated.

The online version of this article includes the following source data and figure supplement(s) for figure 1:

**Source data 1.** Labeled uncropped gel images in *Figure 1*.

**Source data 2.** Raw gel images in *Figure 1*.

**Figure supplement 1.** Validation of primary adipocyte differentiation and overview of MS results.

absolute quantification (iBAQ) value estimates a protein's molar abundance, and summation of iBAQ intensities ($\Sigma$ *iBAQ*) of a sample can estimate total moles of proteins detected (*Shin et al., 2013*). $\Sigma$ iBAQ values closely correlated with in-gel fluorescence (*Figure 1C*), both of which showed highest abundance of secreted proteins in SubQ CM (*Figure 1D*).

We next examined how many of these AHA-labeled proteins are reported or predicted secreted factors. Referencing gene ontology (GO) cellular component terms for 594 genes from 604 detected proteins, a large majority of proteins (413/594, 70%) were annotated to be extracellular (*Figure 1E*). We performed retrospective secretion prediction analysis on 604 proteins, evaluating the proportion of proteins predicted to be secreted by the classical ER/Golgi pathway (SignalP5.0, TMHMM2.0), sequence-based deep learning method for subcellular localization prediction (DeepLoc1.0), glyco-sylphosphatidylinositol (GPI)-anchors (PredGPI), and non-classical routes (SecretomeP2.0) (*Almagro Armenteros et al., 2017*; *Almagro Armenteros et al., 2019*; *Bendtsen et al., 2004*; *Krogh et al., 2001*; *Pierleoni et al., 2008*). Over two-thirds of the proteins (413/604, 68%) were predicted to be secreted via the classical pathway, while non-classical secretion constituted about 10% (62/604) (*Figure 1F*). Overall, 79% of detected proteins (479/604) met at least one criterion of secretion prediction. Notably, more than 20% of identified proteins would not have been predicted to be secreted by these algorithms, highlighting the imperfect nature of in silico predictions and the value of directly measuring secreted protein levels.

In-gel fluorescence analysis showed differential band patterns across the three types of adipocytes, suggesting cell type-specific secretory profiles (*Figure 1C*). Consistent with this observation, principal component analysis (PCA) showed that each cell type formed a distinct cluster (*Figure 1—figure supplement 1F*). Comparing all nine AHA-pulsed samples against each other showed higher correlations within biological replicates ($R>0.97$) than those across different cell types ($0.74<R < 0.91$) (*Figure 1—figure supplement 1G*). For quantitative analysis, we focused on 424 proteins detected in at least 2/3 replicates (*Figure 1—figure supplement 1E*) and compared their Log2-transformed label-free quantification (LFQ) intensities, imputing any missing values with a left-shifted Gaussian distribution. One-way ANOVA yielded 348 proteins with significantly different ($q<0.01$) secretory profiles across cell types, and unbiased clustering was performed based on protein levels in CM (*Figure 1G*). Clusters 1 and 4 contained proteins enriched in CM of Visc and Brown, respectively. Cluster 2 proteins were most abundant in CM of SubQ followed by Visc, both white adipocytes. Cluster 3 was highly secreted by SubQ and Brown adipocytes with proteins such as SLIT2 (*Svensson et al., 2016*), suggesting enrichment of the beige/brown secretome, as well as regulators of immune cell migration and angiogenesis. Cluster 4 comprised the largest number of proteins (125/348), consistent with Brown CM containing the greatest number of unique proteins (*Figure 1H*). Ranking proteins by decreasing order of abundance, we found high levels of many well-described adipocyte-derived factors such as CFD, RETN, ADIPOQ, and RBP4 (*Figure 1I*). Interestingly, many cluster 2 proteins ranked highly in abundance, while cluster 4 proteins skewed towards lower abundance (*Figure 1I*). Hence, while SubQ adipocytes demonstrate high secretory capacity, the brown adipocyte secretome is characterized by a diverse array of proteins, many of which are secreted at lower levels.

We next performed pathway analysis on each of the clusters (*Figure 1J*). Visc CM-enriched cluster 1 was overrepresented with proteins involved in immune response and complement activation, such as C1QA and CFH (*Figure 1K*). Proteins with a role in collagen fibril organization and extracellular

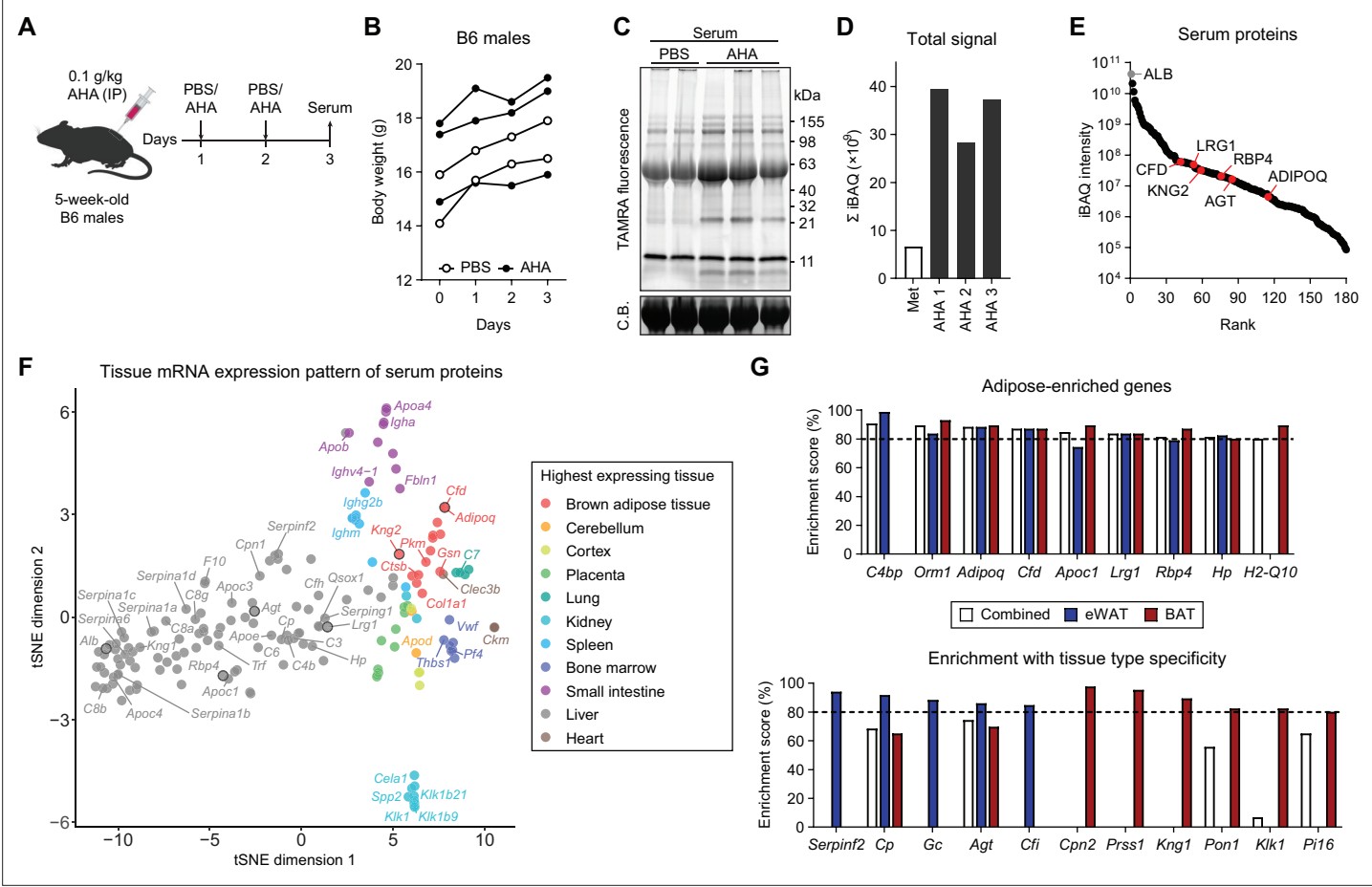

**Figure 2.** Profiling in vivo nascent serum proteome using BONCAT. (**A**) Schematic of AHA IP injections. (**B**) Body weights of mice injected with PBS (n=2) or AHA (n=3). (**C**) In-gel fluorescence analysis of TAMRA-conjugated serum proteins from PBS-pulsed (lanes 1 and 2) or AHA-pulsed (lanes 3–5) mice. C.B., Coomassie Blue. (**D**) Sum of iBAQ intensities (Σ iBAQ) across all quantified proteins in each serum. (**E**) iBAQ intensities of detected serum samples arranged in decreasing order. (**F**) t-SNE clustering of detected serum protein genes based on tissue mRNA levels from the ENCODE/LICR dataset. (**G**) Adipose tissue-enriched serum protein genes and % enrichment scores calculated from Bio-GPS dataset.

The online version of this article includes the following source data for figure 2:

**Source data 1.** Labeled uncropped gel images in *Figure 2*.

**Source data 2.** Raw gel images in *Figure 2*.

matrix formation, such as COL1A1 and SFRP, were enriched in cluster 2 (*Figure 1L*). Cluster 3 (beige/ brown enriched) showed overrepresentation of angiogenesis regulators (*Figure 1M*). Finally, cluster 4 Brown CM-specific proteins were enriched for axon guidance factors such as NRP1 and NGF, consistent with the importance of innervation in thermogenesis (*Figure 1K*; *Chi et al., 2018*; *Chi et al., 2021*; *Wang et al., 2020*; *Zeng et al., 2019*).

## Profiling of the serum proteome in vivo

Previous studies have shown that AHA can be administered in vivo to label tissue proteins (*Calve et al., 2016*; *McClatchy et al., 2015*). However, it has not been tested whether this method can label the nascent serum proteome. We administered 0.1 g/kg/day of AHA IP to chow-fed B6 mice for 2 days (*Figure 2A*) and did not notice any major adverse effects based on body weight (*Figure 2B*). CuAAC conjugation of serum proteins with TAMRA-alkyne and in-gel fluorescence analysis showed increased signal across most bands in the AHA group along with some AHA group-specific fluorescent bands (*Figure 2C*). We used alkyne-agarose beads to enrich the azide-labeled nascent proteome and performed MS analysis. Σ iBAQ showed successful enrichment in animals injected with AHA (*Figure 2D*). Even without dietary Met restriction or depletion of abundant serum proteins, we were

able to identify and quantitate 180 proteins, including classical adipokines such as ADIPOQ, adipsin (CFD), and RBP4 (*Figure 2E*).

Because AHA can be incorporated into the proteome of any tissue, we employed bioinformatic analyses to compare tissue expression levels of genes that encode serum proteins. We cross-referenced 177 genes from 180 detected proteins with publicly available transcriptomic datasets, such as ENCODE (RNA-Seq based) and BioGPS (microarray based) (*Davis et al., 2018*; *Wu et al., 2016*). A t-distributed stochastic neighbor embedding (t-SNE) plot was generated with each protein and its tissue mRNA levels from the ENCODE/LICR dataset to visualize each gene's basal mRNA expression pattern across multiple tissues from lean adult mice (*Figure 2F*). The majority of proteins were most highly expressed by the liver, which comprised the largest cluster (98/177, 55%). BAT, the only adipose tissue profiled in the dataset, formed a cluster of 14 genes (7.9%). As expected, classical adipokines known for highly adipose-specific expression such as adiponectin (*Adipoq*) and adipsin (*Cfd*) belonged to this group, along with a recently described batokine, *Kng2* (*Peyrou et al., 2020*).

BioGPS offers microarray-based transcriptomic data across a much wider variety of mouse tissue and cell types, including eWAT and BAT. We numerically scaled the degree of adipose tissue enrichment for each detected serum protein. Adipose enrichment of a gene was defined as the number of tissues with expression significantly lower than that of eWAT, BAT, or combined. We divided this value by the total number of pairwise comparisons (i.e., total number of tissues - 2 adipose tissues) to obtain the percentage adipose tissue enrichment score. Among the top-enriched genes were well-described adipokines such as *Adipoq*, where 76 of 86 tissues (88.4%) expressed *Adipoq* at significantly lower levels than eWAT and BAT combined (*Figure 2G*). The remaining 10 tissues that did not meet the statistical significance cutoff still expressed *Adipoq* at lower levels compared to eWAT or BAT and included adipose-embedded tissues such as mammary glands. We also identified genes yet to be described as adipokines, such as *H2-Q10* and *Cpn2* (*Figure 2G*). Many genes, such as *Rbp4*, *Agt*, and *Lrg1*, showed high expression in adipose tissues as well as liver. On the t-SNE plot, these genes were grouped with other liver-specific genes, but located closer to the adipose tissue cluster (*Figure 2F*). Still, their percentage enrichment scores were >80%, as few other tissues express those genes.

## LRG1 is secreted by mature adipocytes and increased in obesity

To identify uncharacterized adipokines with a putative role in whole-body metabolism, we prioritized factors (1) detected in adipocyte CM, (2) enriched in adipose tissues with scores >80%, and (3) present in the nascent serum proteome (*Figure 3A*). Intersection analysis between (1) and (2) identified 23 proteins (*Supplementary file 1*), of which 9 proteins were identified in the serum proteome (*Figure 3A*). ADIPOQ, CFD, and RBP4 have already been identified as adipokines, validating our search strategy. Haptoglobin (HP), ceruloplasmin (CP), and angiotensinogen (AGT) have well-characterized biological functions. We therefore focused on LRG1, a protein with relatively unknown metabolic function.

LRG1, or leucine-rich α–2 glycoprotein 1, has been shown to promote angiogenesis by modulating TGF-β signaling (*Wang et al., 2013*), and this process has been implicated in various complications of diabetes (*Hong et al., 2019*; *Liu et al., 2020*; *Zhang et al., 2019*). In 3T3-L1 adipocytes, *Lrg1* expression correlates with adipogenesis (*Birsoy et al., 2011*). In BAT, *Lrg1* expression is dependent on an EHMT1-driven differentiation program (*Ohno et al., 2013*). Recently, whole-body LRG1 loss of function has been reported to reduce obesity and improve metabolic health by reducing hepatosteatosis (*He et al., 2021*). However, the role of adipose-derived LRG1 in whole-body metabolism has not been studied. Consistent with our tissue enrichment analysis, qPCR of mouse tissues showed that *Lrg1* mRNA was mostly expressed in adipose tissues and liver (*Figure 3B*). To determine which cell type(s) within adipose tissue express *Lrg1*, we fractionated eWAT, iWAT, and BAT to separate floating mature adipocytes from the SVF. We confirmed successful enrichment of *Fabp4*, *Pparg2*, and *Adipoq* in the mature adipocyte fraction and *Adgre1* and *Pecam1* in the SVF fraction (*Figure 3—figure supplement 1A*). *Lrg1* was successfully co-enriched in the adipocyte fraction in all three depots (*Figure 3C*). Finally, we cultured primary Visc, SubQ, and Brown adipocytes and confirmed that *Lrg1* mRNA is induced >70-fold in all three cell types during in vitro adipogenesis (*Figure 3D*). Consistent with mRNA data, we found robust levels of LRG1 protein in CM of mature adipocytes, whereas preadipocytes did not secrete detectable LRG1 (*Figure 3E*, quantified in *Figure 3—figure supplement 1B*). Of note, LRG1 protein levels detected by western blot were consistent with MS results (*Figure 3—figure supplement 1C*).

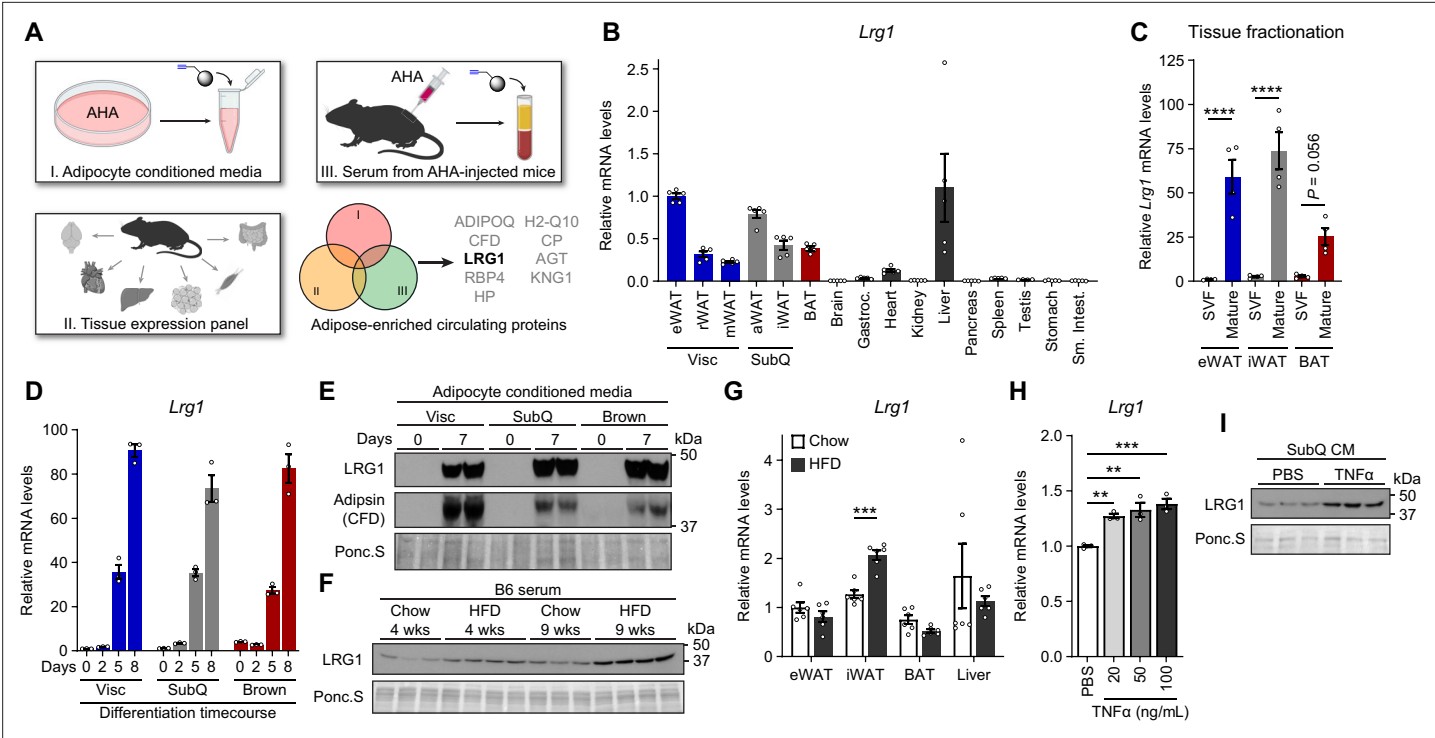

**Figure 3.** LRG1 is secreted by mature adipocytes and increased in obesity. (**A**) Search strategy for identification of novel adipokines with a potential role in whole-body metabolism. (**B**) Relative *Lrg1* mRNA levels across a panel of tissues from 8-week-old male B6 mice. Visc, visceral fat depots; SubQ, subcutaneous fat depots. n=4–5 biological replicates per group. (**C**) Relative *Lrg1* mRNA levels of indicated adipose tissues fractionated into mature adipocytes and SVF. ****p<0.0001 from Šídák post hoc test following two-way ANOVA. n=4 biological replicates per group. (**D**) Relative *Lrg1* mRNA levels during in vitro differentiation of primary SVF into adipocytes. n=3 technical replicates per group. (**E**) Western blot of LRG1 and CFD in CM before (day 0) and after (day 7) in vitro adipogenic differentiation, in technical replicates. (**F**) Serum western blot of LRG1 from male B6 mice on standard chow or HFD for indicated weeks. HFD was started at 6 weeks of age. Each lane represents a biological replicate. (**G**) Relative *Lrg1* mRNA levels of indicated tissues from mice on standard chow or HFD for 15 weeks. HFD was started at 6 weeks of age. ***p<0.001 from two-sided Welch's *t*-test. n=6 biological replicates per group. (**H**) Relative *Lrg1* mRNA levels of primary SubQ adipocytes treated with recombinant TNFα and 850 nM insulin for 6 hr. **p<0.01, ***p<0.001 from Dunnett post hoc test following one-way ANOVA. n=3 technical replicates per group. (**I**) Western blot of LRG1 in CM of primary SubQ adipocytes treated with PBS or 100 ng/mL recombinant TNFα without insulin for 24 hr. Each lane represents a technical replicate. Data are presented as mean ± SEM.

The online version of this article includes the following source data and figure supplement(s) for figure 3:

**Source data 1.** Uncropped gel and western blot images in *Figure 3* and *Figure 3—figure supplement 2*.

**Source data 2.** Raw gel and western blot images in *Figure 3* and *Figure 3—figure supplement 2*.

**Figure supplement 1.** Validation of tissue fractionation and quantification of LRG1 in adipocyte CM.

**Figure supplement 2.** LRG1 gain or loss of function does not affect adipogenesis.

Because *Lrg1* expression correlated with adipogenesis, we next sought to determine if *Lrg1* directly contributes to adipocyte differentiation. LRG1 has been reported to bind cytochrome *c* (Cyt *c*), and this property has been used for depletion of LRG1 by adsorption (*Codina et al., 2010*; *Cummings et al., 2006*). We depleted bovine LRG1 in FBS by incubating with Cyt *c*-conjugated agarose beads and collecting the supernatant. We validated that after 2 rounds of depletion, LRG1 in FBS was no longer detectable, and elution of bound proteins from the Cyt *c*-agarose beads confirmed specific depletion of LRG1 (*Figure 3—figure supplement 2A*). SVF from adipose tissues of LRG1-KO mice (generation of which is described further below) was able to differentiate into mature adipocytes in LRG1-depleted media, and the degree of adipogenesis was largely unaffected by adding back recombinant human LRG1 (rhLRG1), as determined by lipid accumulation and mature adipocyte gene expression (*Figure 3—figure supplement 2B, C*). These results demonstrate that LRG1 does not affect adipogenesis.

To assess whether adipose tissues are a significant contributor to circulating LRG1 levels, we collected serum from chow-fed lean mice or diet-induced obese (DIO) mice on high fat diet (HFD) for 4 or 9 weeks. Serum LRG1 protein levels increased with age and obesity (*Figure 3F*, quantified in *Figure 3—figure supplement 2D*). qPCR of major *Lrg1*-expressing tissues showed significant induction of *Lrg1* mRNA in iWAT of DIO mice, but not in liver (*Figure 3G*). With iWAT expansion in obesity, this induction likely contributes to elevated serum LRG1 levels in this state. Obesity is characterized by chronic low-grade inflammation, with elevated circulating inflammatory cytokines (*Lackey and Olefsky, 2016*). We observed that treatment of primary SubQ adipocytes with recombinant TNFα induced expression of *Lrg1* mRNA (*Figure 3H*) and protein in (*Figure 3I*, quantified in *Figure 3—figure supplement 2E*). Taken together, these results demonstrate that LRG1 is an obesity-induced adipokine.

## LRG1 overexpression improves glucose homeostasis in diet-induced obesity

To explore whether LRG1 as an adipokine affects whole-body energy homeostasis, we used viral vectors to overexpress LRG1 in vivo. Adenovirus encoding eGFP (Ad-eGFP) or C-terminally FLAG-tagged LRG1 (Ad-LRG1-FL) was administered to obese B6 mice on HFD for 10 weeks (*Figure 4A*). Plasma western blot confirmed 44-fold overexpression of LRG1 in the Ad-LRG1-FL group 5 days after infection (*Figure 4B, C*). We observed no difference in body weights between groups (*Figure 4D*), but an insulin tolerance test (ITT) showed that the Ad-LRG1-FL group had a significantly enhanced insulin response (p=0.045) (*Figure 4E*).

To study the longitudinal effects of chronic LRG1 overexpression, we used AAV8, which has tropism for liver and adipose tissue. We administered AAV-eGFP or AAV-LRG1-FL to 6-week-old male B6 mice (*Figure 4F*) and confirmed 12-fold increased plasma LRG1 in the latter (*Figure 4G, H*). Tissue qPCR showed significant *Lrg1* overexpression in the liver, iWAT, and BAT, with the greatest increase in the liver (*Figure 4—figure supplement 1A*). At the protein level, however, eWAT and iWAT demonstrated the highest degree of overexpression (*Figure 4I*, quantified in *Figure 4—figure supplement 1B*). During 3 months of HFD, both groups gained an equivalent amount of weight to around 50 g (*Figure 4J*), with no difference in tissue weights at sacrifice (*Figure 4—figure supplement 1C*). Fasting glucose measurements from 4 to 14 weeks on HFD showed that the AAV-LRG1-FL group had significantly lower fasting glucose levels (p=0.0097) with a dampened peak (LRG1: 225.9±6.5 mg/dL vs. eGFP: 263.7±10.8 mg/dL, mean ± SEM) at week 18 (*Figure 4K*). Glucose and insulin tolerance tests demonstrated that AAV-LRG1-FL mice had markedly improved glucose (p=0.0029) (*Figure 4L*, AUC shown in *Figure 4—figure supplement 1D*) and insulin tolerance (p=0.023) (*Figure 4M*, AOC shown in *Figure 4—figure supplement 1E*). To determine if insulin secretion is affected by LRG1, we performed a glucose-stimulated insulin secretion (GSIS) assay. Glucose tolerance was significantly improved in the AAV-LRG1-FL group (p=0.020) (*Figure 4—figure supplement 1F*) without a significant difference in plasma insulin secretion (*Figure 4—figure supplement 1G*), and plasma non-esterified fatty acid (NEFA) levels were completely suppressed in both conditions (*Figure 4—figure supplement 1G, H*). The homeostatic model assessment of insulin resistance (HOMA-IR) is a widely used model to estimate insulin resistance (*Matthews et al., 1985*). HOMA-IR values trended lower in the LRG1 group, consistent with insulin sensitization (*Figure 4—figure supplement 1I*). To determine whether hepatic gluconeogenesis contributes to the difference in fasting glucose levels, we performed an oral pyruvate tolerance test (PTT) (*Figure 4—figure supplement 1J*) and qPCR of hepatic gluconeogenic gene expression (*Figure 4—figure supplement 1K*) and observed no significant difference between the groups. Taken together, these observations suggest that LRG1 overexpression prevents obesity-related dysregulation of glucose homeostasis by insulin sensitization.

## LRG1 loss of function elevates fasting blood glucose in diet-induced obesity

To test whether LRG1 loss of function affects glucose homeostasis, we generated whole-body LRG1-KO animals in B6 background using CRISPR-Cas9 targeting exon 2 of *Lrg1* (*Figure 4—figure supplement 2A*). This led to a frameshift mutation in *Lrg1* (*Figure 4—figure supplement 2B*) and absence of LRG1 protein in plasma (*Figure 4—figure supplement 2C*). The mouse line was subsequently backcrossed to the parental B6 strain for at least 5 generations to reduce any potential off-target effects. LRG1-KO

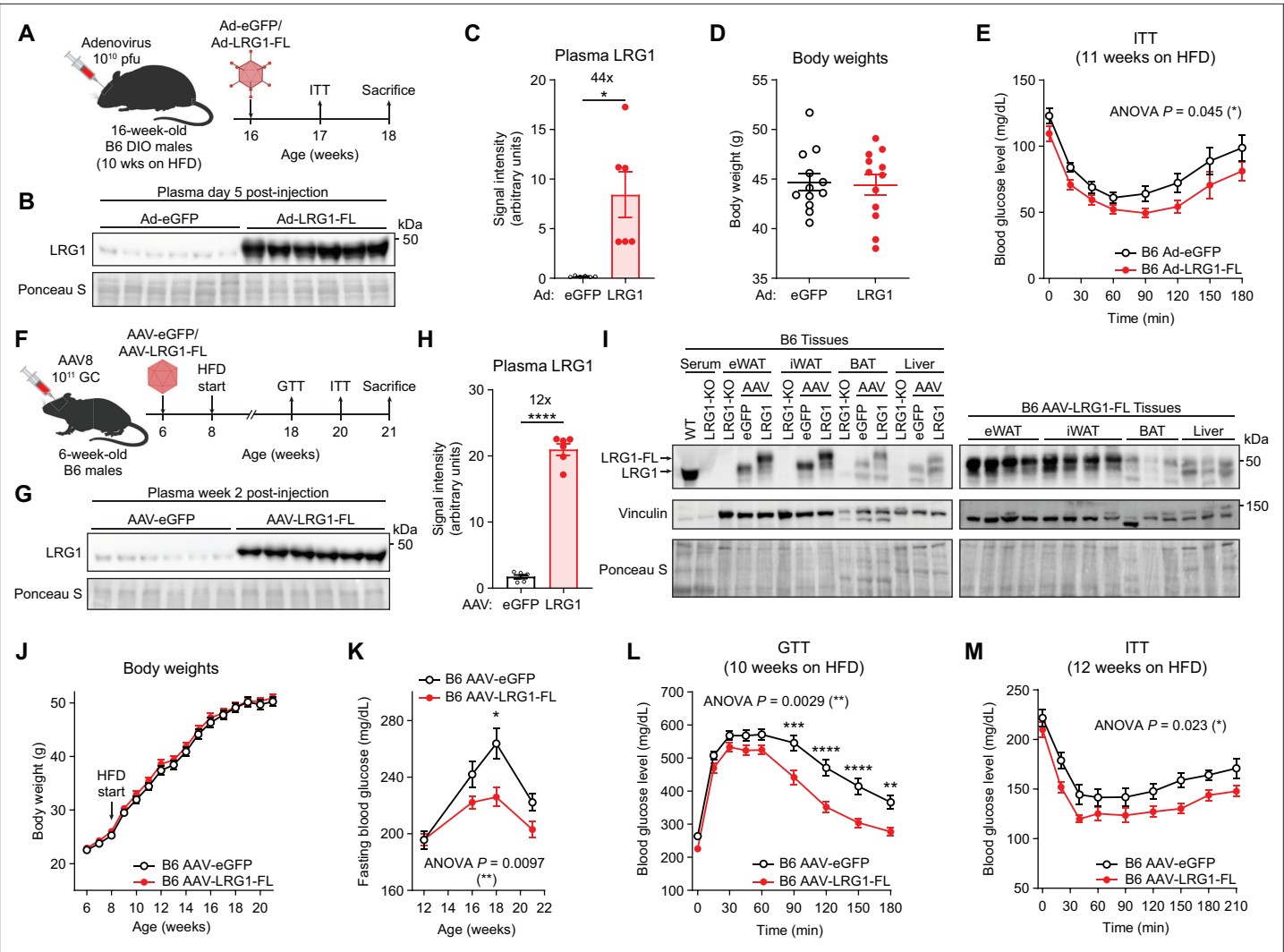

**Figure 4.** LRG1 overexpression improves glucose homeostasis in C57BL/6 J DIO mice. (**A**) Schematic of Ad vector-mediated acute LRG1 overexpression in B6 DIO male mice. (**B**) Plasma western blot of LRG1 5 days after Ad injection. (**C**) Quantification of (**B**). *p<0.05 from two-sided Welch's *t*-test. n=6 per group. (**D**) Body weights of Ad-transduced B6 males at 17 weeks of age (11 weeks on HFD). (**E**) Insulin tolerance test (1.0 U/kg) in Ad-transduced B6 males at 17 weeks of age (11 weeks on HFD). Ad cohort consisted of n=12 per group. (**F**) Schematic of AAV8-mediated chronic LRG1 overexpression in B6 males. AAV injection was performed in 6-week-old B6 males. HFD was started at 8 weeks of age. (**G**) Plasma western blot of LRG1 2 weeks after AAV injection. (**H**) Quantification of (**G**). ****p<0.0001 from two-sided Welch's *t*-test. n=6 biological replicates per group. (**I**) Left: western blot of LRG1 in serum and tissues from WT vs. LRG1-KO and B6 mice transduced with AAV-eGFP vs. AAV-LRG1-FL. Right: western blot of LRG1 in various tissues from AAV-LRG1-FL-transduced B6 mice. (**J**) Body weights of AAV-transduced B6 male mice during HFD challenge. (**K**) 6 hr fasting blood glucose levels of AAV-transduced B6 male mice during HFD feeding. (**L**) Intraperitoneal glucose tolerance test (1.5 g/kg) in AAV-transduced B6 males at 18 weeks of age (10 weeks on HFD). (**M**) Insulin tolerance test (1.5 U/kg) in AAV-transduced B6 males at 20 weeks of age (12 weeks on HFD). AAV cohort consisted of n=12 per group. Data are presented as mean ± SEM. In (**E, K, L, M**), ANOVA p indicates group factor p-values from repeated measures two-way ANOVA. Where indicated, p-values from Šídák post hoc tests are reported. *p<0.05, **p<0.01, ***p<0.001, ****p<0.0001.

The online version of this article includes the following source data and figure supplement(s) for figure 4:

**Source data 1.** Labeled uncropped western blot images in *Figure 4* and *Figure 4—figure supplement 2*.

**Source data 2.** Raw western blot images in *Figure 4* and *Figure 4—figure supplement 2*.

**Figure supplement 1.** Metabolic characterization of AAV-transduced B6 mice.

**Figure supplement 2.** LRG1 loss of function in C57BL/6 J DIO mice elevates fasting blood glucose.

(*Lrg1*⁻/⁻) and WT (*Lrg1*⁺/⁺) littermate controls on HFD showed no difference in body weight (***Figure 4—figure supplement 2D, E***) or tissue weights at sacrifice (***Figure 4—figure supplement 2F***). Interestingly, male LRG1-KO animals demonstrated significantly higher fasting glucose levels compared to WT littermates throughout the HFD challenge (p=0.010) with a higher peak (KO: 261.6±11.3 mg/dL vs. eGFP: 225.0±10.9 mg/dL, mean ± SEM) at week 12 on HFD (***Figure 4—figure supplement 2G***). Therefore, fasting blood glucose levels are reciprocally regulated by LRG1 gain and loss of function. At the time points tested, however, GTT and ITT did not show significant differences between the genotypes (***Figure 4—figure supplement 2H–K***).

## LRG1 overexpression delays diabetic phenotype and promotes WAT expansion in *db/db* mice

While B6 mice develop severe obesity upon HFD feeding, they demonstrate only transient and mild hyperglycemia with moderate insulin resistance (***Winzell and Ahrén, 2004***; ***Kleinert et al., 2018***). Because LRG1 overexpression in B6 animals mitigated hyperglycemia, we explored whether LRG1 can improve glucose homeostasis in C57BLKS/J-*Lepr^db/db^* (*db/db*) mice, a more extreme model of obesity-related type 2 diabetes. Due to a leptin receptor (*Lepr*) mutation and genetic background, *db/db* animals demonstrate hyperphagia and early-onset obesity, along with profound hyperglycemia and hyperinsulinemia (***Kleinert et al., 2018***). We confirmed that obesity and hyperglycemia over 400 mg/dL develop in *db/db* animals as early as 7 weeks of age, while littermate *misty* mice with a WT *Lepr* gene (*m/m*) maintain fasting glucose levels below 200 mg/dL (***Figure 5—figure supplement 1A, B***). Similar to B6 DIO mice, *db/db* mice showed higher circulating LRG1 levels compared to lean littermates (***Figure 5—figure supplement 1C, D***).

We administered AAV-eGFP or AAV-LRG1-FL to *db/db* mice at 4 weeks of age, before development of severe hyperglycemia (***Figure 5A***). In LRG1-overexpressing *db/db* mice, we confirmed 4.3-fold increased circulating LRG1 (***Figure 5B, C***) and that the adipose tissues were the major sites of *Lrg1* overexpression (***Figure 5—figure supplement 1E, F***). Starting 2 weeks post-injection, LRG1-overexpressing animals demonstrated accelerated weight gain, such that by 10 weeks of age the LRG1 group weighed 19.6% more (p=0.004) than eGFP controls (***Figure 5D***). Concomitantly, the AAV-LRG1-FL group showed delayed onset of hyperglycemia. At week 6, we observed frank hyperglycemia in eGFP animals, whereas glucose levels in the LRG1 group were 33.6% lower (eGFP: 399.5±31.7 mg/dL vs. LRG1: 265.4±19.6 mg/dL, mean ± SEM; p=0.0020) (***Figure 5E***). Fasting plasma insulin levels were almost halved in the LRG1 group (eGFP: 12.5±1.3 ng/mL vs. LRG1: 6.3±0.6 ng/mL, mean ± SEM; p=0.0008) (***Figure 5F***). HOMA-IR was significantly lower in LRG1-overexpressing mice (***Figure 5—figure supplement 1G***). Consistent with these findings, ITT profiles significantly differed between the two groups (p=0.019) (***Figure 5G***), largely due to differences in basal glucose levels (***Figure 5—figure supplement 1H***). By 8 weeks of age, body weights continued to diverge between the groups, but we no longer observed significant differences in fasting blood glucose or plasma insulin concentrations (***Figure 5D–F***).

To further characterize the metabolic effects of LRG1 overexpression in *db/db* animals, we performed indirect calorimetry between 6 and 7 weeks of age (***Figure 5A***). Measurement of total energy expenditure (EE) across 5 days following acclimation (***Figure 5H***) and regression analysis against body weight using a generalized linear model using CalR (***Mina et al., 2018***) showed that there is a significant difference in EE between the two groups (p=0.0010) as well as a significant interaction effect between EE and body weight (p=0.0014) (***Figure 5I***). This suggests that LRG1 overexpression leads to a greater increase in EE per unit increase in body weight. Respiratory exchange ratio (RER) was higher in the LRG1 group compared to eGFP controls (eGFP: 0.89±0.013 vs. LRG1: 0.93±0.014, mean ± SEM; p=0.046), suggesting a fuel preference for carbohydrates (***Figure 5J, K***). The eGFP group showed greater locomotor activity than the LRG1 group (eGFP: 234±17.5 vs. LRG1: 185±9.27, mean ± SEM; p=0.027), especially during the dark cycle (***Figure 5L, M***). Finally, LRG1-overexpressing animals had significantly greater food intake, consuming approximately 14% more food each day (p=0.039) (***Figure 5N***).

Tissue weight measurements revealed that LRG1-overexpressing animals had accelerated gain of eWAT (p<0.0001) and iWAT (p=0.0001) mass, such that by week 10, their eWAT and iWAT were 53% and 40% heavier, respectively, than those of eGFP controls (***Figure 5O, P***). We observed no difference in weights of BAT, liver, or gastrocnemius muscle (***Figure 5O***). To further determine if LRG1

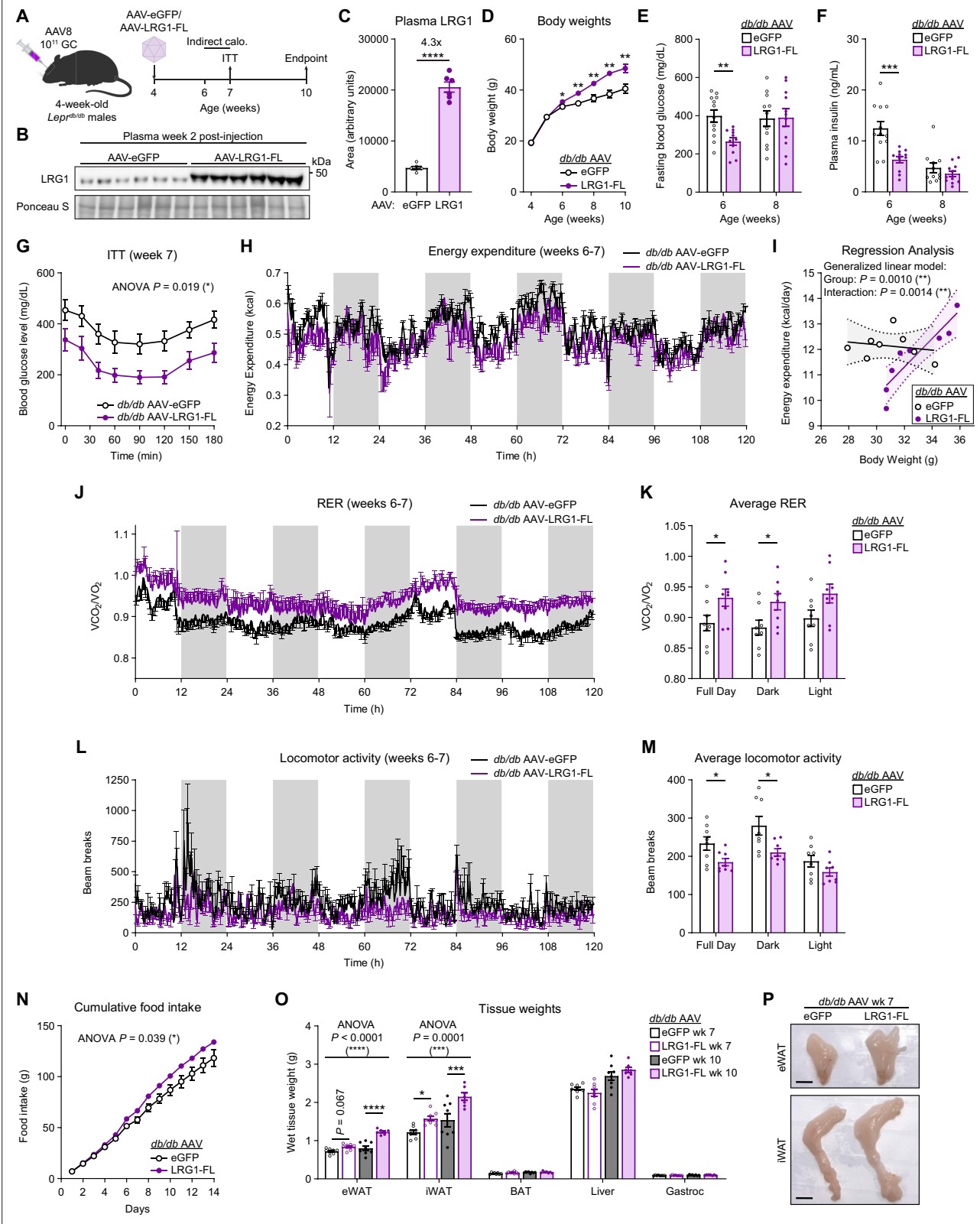

**Figure 5.** LRG1 overexpression in C57BLKS/J-*Lepr*[db/db] mice delays onset of diabetic phenotype and promotes WAT expansion. (**A**) Schematic of AAV8-mediated LRG1 overexpression in *db/db* males. (**B**) Plasma western blot of LRG1 2 weeks after AAV injection. (**C**) Quantification of (**B**). ****p<0.0001 from two-sided Welch's *t*-test. n=6 per group. (**D**) Body weights of AAV-transduced *db/db* male mice on standard chow diet. *p<0.05, **p<0.01 from two-sided Welch's *t*-test. n=16 per group for weeks 4–6; n=7–8 per group for weeks 7–10. (**E**) 6 hr fasting blood glucose levels of AAV-transduced *db/db*

*Figure 5 continued on next page*

*Figure 5 continued*

male mice at 6 and 8 weeks of age. **p<0.01 from two-sided Welch's *t*-test. n=11–12 per group. (**F**) Plasma insulin levels from (**D**). ***p<0.001 from two-sided Welch's *t*-test. (**G**) Insulin tolerance test (2.0 U/kg) in AAV-transduced 7-week-old *db/db* male mice. ANOVA p indicates group factor p-value from repeated measures two-way ANOVA. *p<0.05. n=7–8 per group. (**H**) Energy expenditure (EE) of AAV-transduced *db/db* male mice measured by indirect calorimetry over the course of 5 days (120 hr). Shaded regions indicate dark cycles. (**I**) Regression analysis of EE vs. body weight from (**H**). (**J**) Respiratory exchange ratio (RER) of AAV-transduced *db/db* male mice in (**H**). (**K**) Daily average of (**J**) *p<0.05 from one-way ANOVA. (**L**) Locomotor activity of AAV-transduced *db/db* male mice in (**H**). (**M**) Daily average of (**L**) *p<0.05 from one-way ANOVA. (**N**) Cumulative daily food intake over the course of 14 days. *p<0.05 from one-way ANOVA. Indirect calorimetry and food intake measurements were performed with n=8 per group. (**O**) Weights of dissected tissues from AAV-transduced *db/db* male mice at 7 and 10 weeks of age. ANOVA p indicates group factor p-values from two-way mixed effects ANOVA. Where indicated, p-values from Šídák post hoc tests reported. *p<0.05, ***p<0.001, ****p<0.0001. n=7–8 per group. (**P**) eWAT and iWAT of AAV-transduced *db/db* male mice at 7 weeks of age. Scale bars indicate 1 cm. Data are presented as mean ± SEM.

The online version of this article includes the following source data and figure supplement(s) for figure 5:

**Source data 1.** Labeled uncropped western blot images in *Figure 5* and *Figure 5—figure supplement 1*.

**Source data 2.** Raw western blot images in *Figure 5* and *Figure 5—figure supplement 1*.

**Figure supplement 1.** Metabolic characterization of *db/db* vs. *m/m* mice and AAV-transduced *db/db* mice.

affects the insulin signaling pathway, we measured IRS-1 protein levels in eWAT and liver. The LRG1 group showed increased IRS-1 protein, especially in eWAT (*Figure 5—figure supplement 1I and J*). qPCR analysis of hepatic gluconeogenic gene expression showed no significant difference except for *Pck1*, which was higher in the LRG1 group (*Figure 5—figure supplement 1K*). These results reveal that LRG1 overexpression elicits a constellation of metabolic effects in *db/db* animals that result in improved insulin sensitivity and WAT expansion-driven weight gain.

## LRG1 suppresses obesity-associated systemic inflammation

Based on the accelerated eWAT and iWAT expansion in LRG1-overexpressing *db/db* animals, we hypothesized that WAT could be a target of LRG1 action. We analyzed paraffin-embedded, hematoxylin and eosin (H&E) stained tissue sections and found that eWAT from B6 animals sacrificed at 21 weeks of age (13 weeks on HFD) was characterized by accumulation of macrophages forming crown-like structures (CLS) (*Figure 6A*). The distal portion of eWAT was especially susceptible to CLS formation in eGFP controls, while the same region in LRG1-overexpressing animals displayed an 82% reduction in CLS number (p<0.0001) (*Figure 6A, B*). B6 iWAT contained fewer CLS compared to eWAT and did not show major morphological differences between groups (*Figure 6—figure supplement 1A, B*). In *db/db* eWAT, we observed significantly reduced CLS in the LRG1 group by 81% at week 7 (p=0.0001) and 85% at week 10 (p<0.0001) (*Figure 6C, D*). CLS in *db/db* iWAT from LRG1-overexpressing animals were reduced by 83% at week 7 (p=0.0043) and 88% at week 10 (p<0.0001) (*Figure 6C, E*). Quantification of adipocyte cross-sectional areas in *db/db* eWAT slides showed significantly larger adipocyte size in the LRG1 group, especially at week 7 (p=0.0052) (*Figure 6—figure supplement 1C*).

Obesity is associated with non-alcoholic fatty liver disease (NAFLD), characterized by hepatic steatosis with or without inflammation (*Farrell et al., 2019*). While HFD-fed B6 mice rarely demonstrate liver injury or inflammation, mild necroinflammation can be observed in *db/db* liver as early as 1 month of age (*Trak-Smayra et al., 2011*). Liver sections from the B6 cohort showed a similar degree of steatosis in both groups, without any inflammatory lesions (*Figure 6—figure supplement 1D*). In the *db/db* cohort, both eGFP and LRG1 showed no statistically significant differences in the degree of hepatosteatosis (*Figure 6—figure supplement 1E*), but inflammatory foci were found only in eGFP-expressing animals (*Figure 6C*).

Inflammation is a key link between obesity and insulin resistance (*Saltiel and Olefsky, 2017*). We hypothesized that LRG1 may mediate its insulin sensitizing effect via attenuation of inflammation in susceptible organs. We performed RNA-Seq analysis on eWAT from AAV-treated *db/db* animals harvested at week 7 (midpoint) and 10 (endpoint). We performed differential gene expression analysis between the eGFP and LRG1 groups at midpoint and subjected the list of significant genes (p<0.01) to GO gene-set enrichment analysis (GSEA). All of the top 20 differentially regulated pathways showed highly significant enrichment (p=0.0067) and were immune-related, including leukocyte activation, innate immune response, and cytokine production (*Figure 6F*). The enrichment score for each of these pathways was negative in the LRG1 group, indicating down-regulation of inflammatory processes in

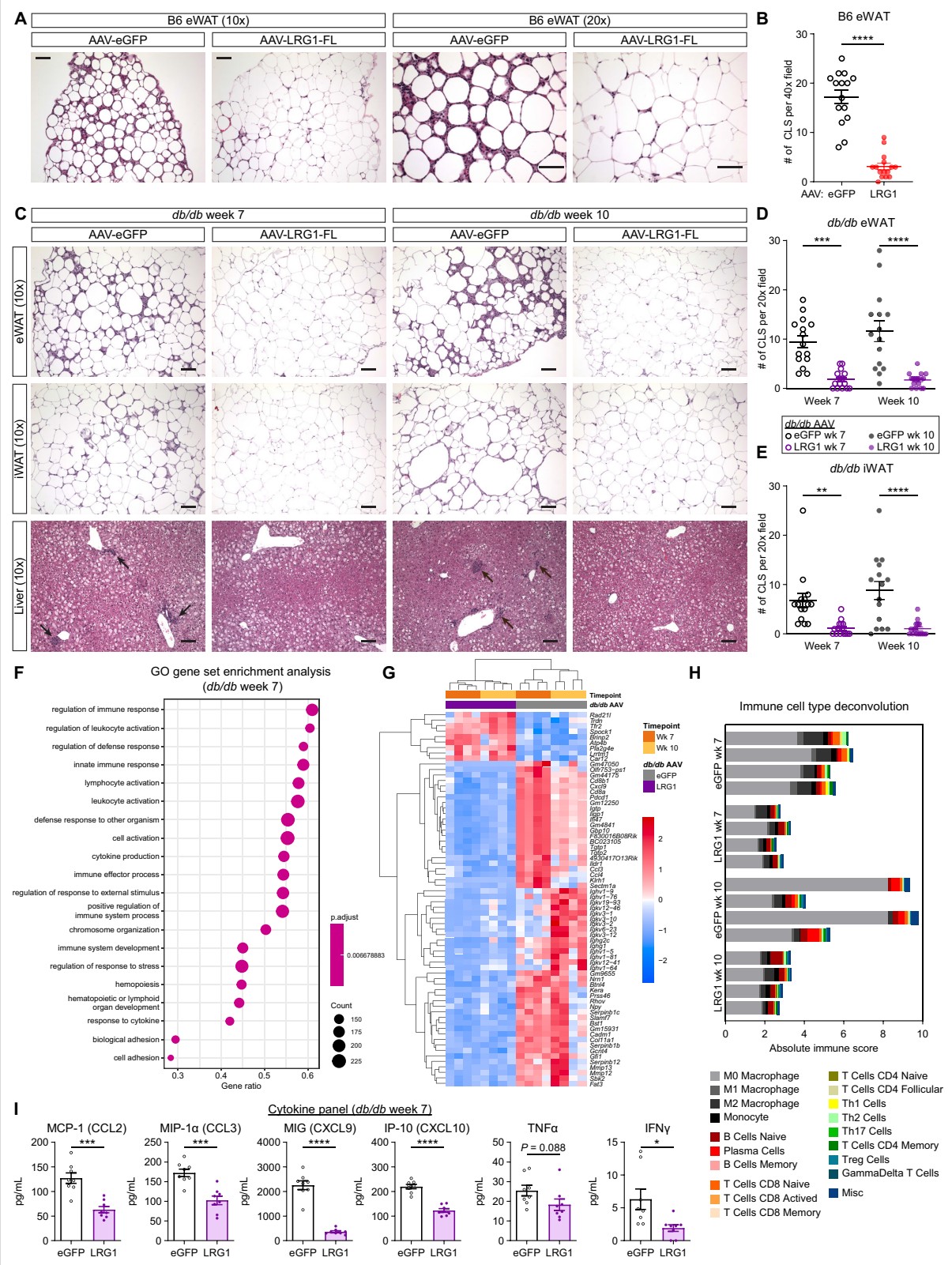

**Figure 6.** LRG1 suppresses obesity-associated systemic inflammation. (**A**) Representative images from H&E-stained eWAT sections from B6 DIO mice transduced with AAV-eGFP or AAV-LRG1-FL under 10 x (left) and 20 x (right) objectives. Scale bars indicate 100 μm. (**B**) Quantification of CLS from (**A**). ****p<0.0001 from two-sided Welch's *t*-test. 5 fields under a 40 x objective from three animals were used for quantification (n=15 per group). (**C**) Representative images from H&E-stained eWAT, iWAT, and liver sections from *db/db* mice transduced with AAV-eGFP or AAV-LRG1-FL. Arrows indicate

*Figure 6 continued on next page*

*Figure 6 continued*

liver inflammatory foci. Scale bars indicate 100 μm. (**D, E**) Quantification of CLS in eWAT (**D**) and iWAT (**E**) from (**C**). Šídák post hoc test results from two-way ANOVA are indicated. **p<0.01, ***p<0.001, ****p<0.0001. Five fields under a 20 x objective from three animals were used for quantification (n=15 per group). (**F**) Top 20 enriched GO BP pathways from GSEA of significantly differentially expressed genes between *db/db*-LRG1 and *db/db*-eGFP eWAT at 7 weeks of age. (**G**) Heatmap of 68 differentially regulated genes (Log₂FC >2 or < –2 and adjusted p<0.01) between *db/db*-LRG1 and *db/db*-eGFP eWAT at both 7 and 10 weeks of age. (**H**) Absolute scores from immune cell type deconvolution analysis of eWAT transcriptomes using CIBERSORTx. (**I**) Quantification of serum chemokine/cytokine levels in *db/db*-LRG1 and *db/db*-eGFP at 7 weeks of age. *p<0.05, ***p<0.001, ****p<0.0001 from two-sided Welch's *t*-test. n=8 biological replicates per group. Data are presented as mean ± SEM.

The online version of this article includes the following figure supplement(s) for figure 6:

**Figure supplement 1.** Analysis of inflammatory phenotypes with LRG1 gain of function.

these animals. GSEA of significantly regulated genes at the 10-week endpoint yielded similar down-regulation of immune-related pathways in the LRG1 group (*Figure 6—figure supplement 1F*). To visualize which genes are most significantly differentially regulated, we plotted a heatmap of 68 genes that showed significant difference (adjusted *P*<0.01) with Log2 fold-change of >2 or <-2 between the two groups at both time points. Consistent with the pathway analysis, the LRG1 group showed significant down-regulation of cytokines and chemokines such as *Ccl3*, *Ccl4*, and *Cxcl9*; metalloproteinases such as *Mmp12* and *Mmp13* known to be highly expressed by macrophages; and various immuno-globulin subunit genes (*Figure 6G*).

Our histological analysis demonstrated significantly reduced CLS in eWAT of LRG1-overexpressing animals. Deconvolution algorithms such as CIBERSORTx allow estimation of cell populations from bulk RNA-Seq datasets (*Newman et al., 2019*). We performed CIBERSORTx analysis on the expression dataset to gain further insight into differences in immune cell populations. CIBERSORTx estimated that the LRG1 group contains fewer immune cells (*Figure 6H*). Macrophages were predicted to constitute the majority of immune cells in every eWAT sample analyzed, and the LRG1 group demonstrated a lower absolute quantity of macrophages (*Figure 6H*), without affecting their relative proportions (*Figure 6—figure supplement 1G*). Many of the differentially regulated genes between eGFP- and LRG1-overexpressing animals encode chemokines and cytokines. We performed a multiplex cytokine assay to assess whether these differences are reflected in serum levels, and at midpoint (week 7), the LRG1 group showed 28–86% reduction of circulating chemokines such as MCP-1, MIP-1α, MIG, and IP-10 and cytokines such as TNFα and IFNγ (*Figure 6I*). Many of these differences subsided by week 10, mostly due to a reduction of cytokine levels in the eGFP group (*Figure 6—figure supplement 1H*). Taken together, these results suggest that LRG1 overexpression attenuated pro-inflammatory processes associated with obesity.

## LRG1 binds extracellular cytochrome *c* and blocks its pro-inflammatory effect on macrophages

LRG1 contains leucine-rich repeat (LRR) domains, which form a structural framework for protein-protein interactions. We hypothesized that LRG1's immunomodulatory function is mediated by protein-protein interactions. LRG1 has been reported to bind cytochrome *c* (Cyt *c*), a mitochondrial protein (*Cummings et al., 2006*). In addition to its role in the respiratory chain and intrinsic apoptosis pathway, Cyt *c* is released into the extracellular space following cell death (*Jemmerson et al., 2002*; *Renz et al., 2001*) and mediates pro-inflammatory signals as a damage-associated molecular pattern (DAMP) (*Grazioli and Pugin, 2018*). Binding between LRG1 and extracellular Cyt *c* has been shown to reduce apoptosis of lymphocytes and cancer cells in vitro (*Codina et al., 2010*; *Jemmerson et al., 2021*). However, the in vivo biological context and effect of this interaction have remained unknown. Adipocyte death in obesity is a key event promoting macrophage infiltration and WAT inflammation (*Cinti et al., 2005*), but the exact triggers of metabolic inflammation remain unidentified. We explored whether extracellular Cyt *c* released by dead/dying adipocytes could be a key mediator of macrophage recruitment/activation, and if LRG1 in turn modulates the pro-inflammatory action of Cyt *c*.

We first examined whether LRG1 binds extracellular Cyt *c* in the circulation. We utilized the C-terminal FLAG-tag of overexpressed LRG1 and α-FLAG antibody-conjugated beads to co-immuno-precipitate (IP) LRG1-FL and interacting partners. In the serum of *db/db* animals transduced with AAV-LRG1-FL, we were able to successfully co-IP both LRG1-FL and Cyt *c* from the serum, indicating the two proteins indeed circulate as a complex (*Figure 7A*). While elevated Cyt *c* levels in circulation

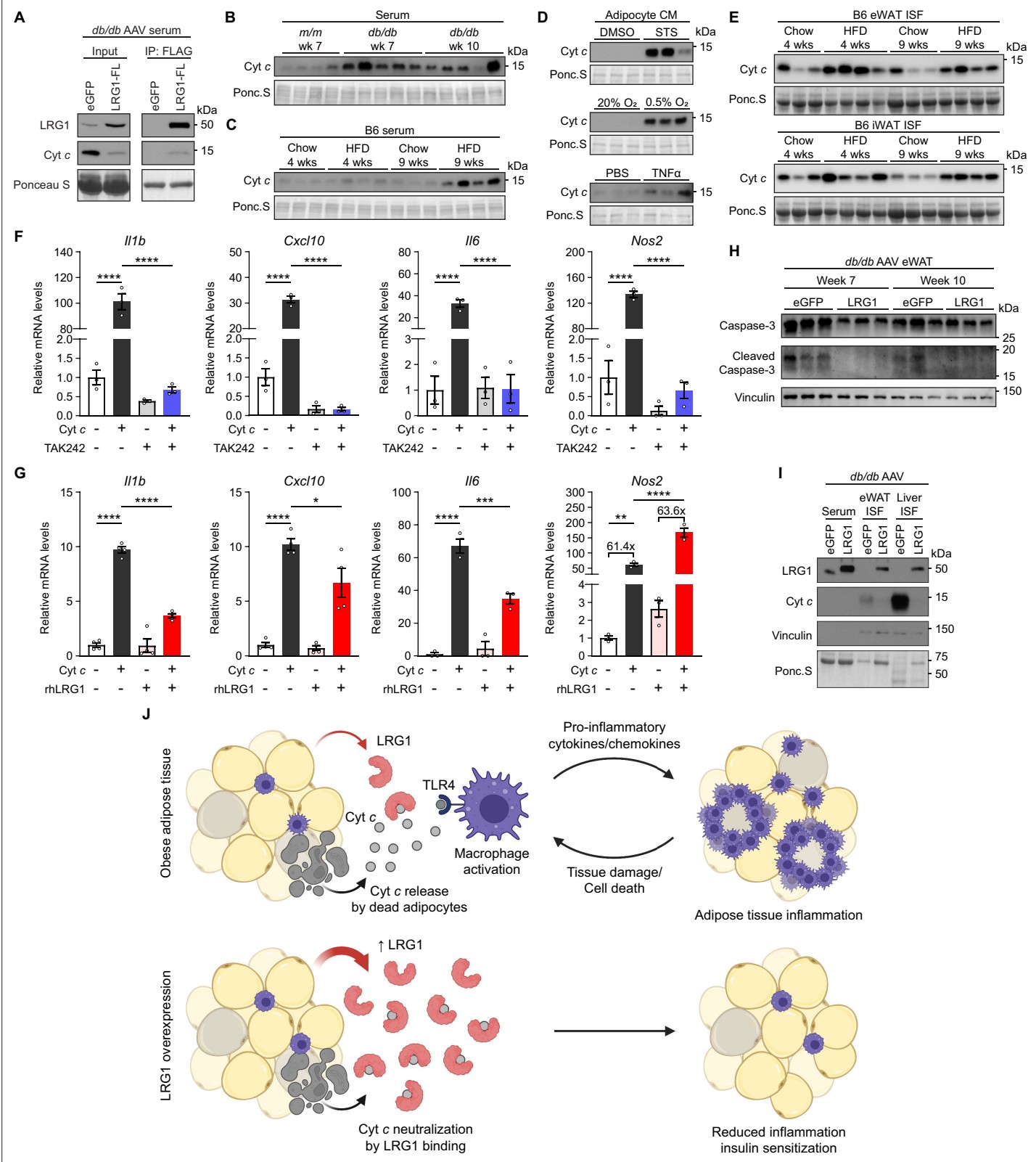

**Figure 7.** LRG1 binds extracellular cytochrome *c* and blocks its pro-inflammatory effect on macrophages. (**A**) α-FLAG co-IP of C-terminally FLAG-tagged LRG1 and Cyt *c* from the serum of AAV-transduced *db/db* mice at 7 weeks of age. (**B**) Western blot of Cyt *c* in the serum of *db/db* or littermate *m/m* mice at 7 or 10 weeks of age. (**C**) Western blot of Cyt *c* in the serum of B6 mice on standard chow or HFD for indicated weeks. HFD was started at 6 weeks of age. Developed from the same membrane in *Figure 3F*. (**D**) Western blot of Cyt *c* in the CM of primary SubQ adipocytes under conditions

*Figure 7 continued on next page*

*Figure 7 continued*

promoting cell death. SubQ cells were treated with 1 µM staurosporine (STS) or DMSO for 24 hr (top); placed in normoxic or hypoxic chambers for 24 hr (middle); or treated with 100 ng/mL recombinant TNFα or PBS without insulin for 24 hr. Developed from the same membrane in *Figure 3I* (bottom). (**E**) Western blot of Cyt *c* in the insterstitial fluid (ISF) of eWAT and iWAT from B6 mice on standard chow or HFD for indicated weeks. (**F**) Relative mRNA levels of a panel of pro-inflammatory genes in BMDMs in response to 30 µg/mL equine Cyt *c* or 10 µM TAK-242, a TLR4 inhibitor. BMDMs were treated with Cyt *c* or PBS for 6 hr following 1 hr pre-incubation with TAK-242 or DMSO. ****p<0.0001 from Tukey post hoc test results following two-way ANOVA. n=3 technical replicates per group. (**G**) Relative mRNA levels of a panel of pro-inflammatory genes in BMDMs in response to 20 µg/mL equine Cyt *c* and 50 µg/mL recombinant human LRG1. Indicated reagents or PBS were rotated for 1 hr at RT prior to BMDM treatment for 6 hr. *p<0.05, **p<0.01, ***p<0.001, ****p<0.0001 from Tukey post hoc test results following two-way ANOVA. n=3–4 technical replicates per group. (**H**) Western blot of intact and cleaved Caspase-3 in eWAT of AAV-transduced *db/db* mice. (**I**) Western blot of LRG1 and Cyt *c* in the serum and ISF of *db/db* mice at 10 weeks of age. Vinculin was used to assess tissue leakage during sample preparation. (**J**) Schematic depicting proposed mechanism. Data are presented as mean ± SEM.

The online version of this article includes the following source data and figure supplement(s) for figure 7:

**Source data 1.** Labeled uncropped western blot images in *Figure 7* and *Figure 7—figure supplement 1*.

**Source data 2.** Raw western blot images in *Figure 7* and *Figure 7—figure supplement 1*.

**Figure supplement 1.** Analysis of LRG1-Cyt *c* interaction.

have been observed in conditions involving cell death or inflammation (*Adachi et al., 2004*; *Alleyne et al., 2001*; *Barczyk et al., 2005*; *Ben-Ari et al., 2003*), it is not known whether obesity is associated with increased serum Cyt *c* levels. Western blot analysis of serum revealed that Cyt *c* is increased in *db/db* animals compared to lean littermates (*Figure 7B*, quantified in *Figure 7—figure supplement 1A*). Interestingly, extracellular Cyt *c* levels trended higher at 7 than at 10 weeks of age (*Figure 7B*, quantified in *Figure 7—figure supplement 1A*), correlating with the time point when LRG1 over-expression has a potent physiological effect. In B6 mice, HFD led to increased serum Cyt *c* levels (*Figure 7C*, quantified in *Figure 7—figure supplement 1B*), and the trend continued at 15 weeks on HFD (*Figure 7—figure supplement 1C, D*). To test if dying adipocytes contribute to circulating Cyt *c*, we subjected primary SubQ adipocytes to conditions that induce apoptosis or necrosis. In CM of SubQ adipocytes treated with staurosporine (STS) for 24 hr, we detected a robust increase in Cyt *c* compared to vehicle controls (*Figure 7D*, quantified in *Figure 7—figure supplement 1E*). CM collected from adipocytes incubated in hypoxic conditions or treated with TNFα also showed increased Cyt *c* compared to normoxic or vehicle controls, respectively (*Figure 7D*, quantified in *Figure 7—figure supplement 1E*). We next collected interstitial fluid (ISF) from eWAT and iWAT of B6 mice and found higher Cyt *c* levels in HFD-fed mice (*Figure 7E*, quantified in *Figure 7—figure supplement 1F*), suggesting adipose tissues are a source of elevated circulating Cyt *c* in DIO mice.

Extracellular Cyt *c* has been shown to exert a pro-inflammatory effect by acting on the toll-like receptor 4 (TLR4)-mediated innate immune signaling pathway in astrocytes (*Wenzel et al., 2019*). Because circulating monocytes contribute to adipose tissue macrophages in obesity (*Oh et al., 2012*), we tested whether bone-marrow derived macrophages (BMDMs) respond to extracellular Cyt *c* by upregulating pro-inflammatory genes. Treating BMDMs with horse Cyt *c* led to >30-fold induction of lipopolysaccharide (LPS)-responsive pro-inflammatory genes such as *Il1b*, *Cxcl10*, *Il6*, and *Nos2* (p<0.0001), while pre-treatment of macrophages with a small molecule TLR4 inhibitor, TAK-242, prevented this induction (*Figure 7F*). These data confirm that Cyt *c* in the extracellular space acts as a DAMP that activates innate immune signaling and polarizes macrophages into a more pro-inflammatory state. We then explored how co-treatment of Cyt *c* and recombinant human LRG1 (rhLRG1) affects Cyt *c*'s pro-inflammatory effect on macrophages. Prior to treatment, media containing Cyt *c*, rhLRG1, or both were incubated for one hour to allow protein-protein interactions to occur. Compared to single treatment, co-treatment with rhLRG1 significantly dampened Cyt *c*-mediated induction of *Il1b* (P<0.0001), *Cxcl10* (P=0.024), and *Il6* (P=0.0008), while *Nos2* induction was not affected by co-treatment with rhLRG1 (*Figure 7G*). Interestingly, pre-treatment of BMDMs with rhLRG1 followed by addition of Cyt *c* did not attenuate the latter's pro-inflammatory effect, suggesting that Cyt *c*-LRG1 complex formation is necessary for LRG1's modulatory effect (*Figure 7—figure supplement 1G*). Expression of alternatively activated (M2) macrophage markers was not affected by co-treatment with Cyt *c* and rhLRG1 (*Figure 7—figure supplement 1H*). We next explored if this immunomodulatory role of LRG1 can in turn reduce adipose tissue damage. Western blot of *db/db* eWAT showed reduction of cleaved Caspase-3 in the LRG1 group (*Figure 7H*), suggesting reduction of apoptotic cells with

LRG1 overexpression. We also observed decreased release of Cyt *c* into the ISF of eWAT and liver (*Figure 7I*), consistent with lower circulating Cyt *c* detected in the serum (*Figure 7A*). Taken together, these results suggest LRG1 can modulate the pro-inflammatory gene expression responses induced by Cyt *c* and subsequently reduce tissue damage and cell death.

Cyt *c* is a small (12.5 kDa), globular protein with a short half-life in circulation (*Radhakrishnan et al., 2007*). Small proteins such as Cyt *c* are expected to be rapidly cleared via glomerular filtration in the kidney. With the tight binding interaction between LRG1 and Cyt *c*, we hypothesized that much of extracellular Cyt *c* circulates as a complex with LRG1. To assess whether this interaction affects Cyt *c* clearance, we performed retroorbital injection of horse Cyt *c* into WT and LRG1-KO mice and collected blood at time points from 5 min to 6 hr post-injection. Plasma western blot showed that LRG1-KO animals almost completely cleared Cyt *c* by 2 hr, whereas in WT animals, an initial decrease in Cyt *c* was followed by nearly constant levels between 1 and 6 hr post-injection (*Figure 7—figure supplement 1I*). Fitting one-phase decay functions to relative Cyt *c* intensities showed that while both WT and LRG1-KO curves have similar half-lives (WT: 16.10 min vs. KO: 17.32 min), the WT trace plateaus at 0.2791, whereas the LRG1-KO curve approaches zero (*Figure 7—figure supplement 1J, K*). Concomitantly, appearance of Cyt *c* in the urine was observed as soon as 30 min after the injection, demonstrating that clearance of excess Cyt *c* is indeed through the kidneys (*Figure 7—figure supplement 1L, M*). Urine Ponceau S staining showed that detected proteins were all below 25 kDa, the apparent size limit for urinary excretion (*Figure 7—figure supplement 1L*). We reasoned that in LRG1-KO animals, Cyt *c* circulates as an unbound, free form that can be rapidly excreted by glomerular filtration, whereas the presence of LRG1 in WT animals prevents a portion of Cyt *c* from excretion due to formation of a complex larger than the size cutoff for glomerular filtration. Indeed, endogenous Cyt *c* was virtually undetectable in the serum of HFD-fed LRG1-KO mice (*Figure 7—figure supplement 1N*). Taken together, these data directly demonstrate that Cyt *c* is a native ligand of LRG1, and that this interaction suppresses the pro-inflammatory effect of extracellular Cyt *c* and cell death (*Figure 7J*).

## Discussion

We applied a chemoproteomic technology to comprehensively profile the secretome of three major types of murine primary adipocytes, revealing unique secretory profiles of each cell type with differential functional enrichment. This method was also applied in vivo to profile the nascent serum proteome, and bioinformatic analysis suggested that adipose tissue is an important contributor to the serum proteome. The intersection of the adipocyte CM and serum proteome included classical adipokines such as adiponectin (*Wang and Scherer, 2016*), adipsin (*Lo et al., 2014*; *Gómez-Banoy et al., 2019*), and RBP4 (*Yang et al., 2005*), all of which have important roles in whole-body energy homeostasis. Leptin was not detected in our dataset, as has been the case in most other MS-based approaches (*Ali Khan et al., 2018*). This is most likely due to its very low expression in in vitro-differentiated adipocytes (*MacDougald et al., 1995*; *O'Connor et al., 2021*). LRG1 shares a similar expression signature with these adipokines, but its metabolic function has not been characterized. We demonstrate, using two types of viral vectors and two different mouse models of obesity, a novel metabolic role for adipose-derived LRG1 as a regulator of glucose homeostasis by promoting insulin sensitization. This insulin-sensitizing effect in LRG1 gain of function is associated with a dramatic reduction in systemic inflammation and with LRG1's ability to bind Cyt *c* and modulate its pro-inflammatory effect as a DAMP.

There have been various other studies that used MS to profile the secretome of adipocytes (*Ali Khan et al., 2018*; *Deshmukh et al., 2019*; *Lehr et al., 2012*; *Roca-Rivada et al., 2015*; *Rosenow et al., 2010*; *Zhou et al., 2009*). Comparison of our dataset with those from two most recent publications (*Ali Khan et al., 2018*; *Deshmukh et al., 2019*) highlights key advances with our approach as well as potential caveats (*Supplementary file 2*). Our dataset stands out for its use of all three major types of adipocytes including Visc cells, which have important implications for cardiometabolic complications of obesity (*Fox et al., 2007*). While the total number of secreted proteins in our dataset may be lower than the other two studies (604 vs 1337 and 1866), the number of known secreted proteins is higher (472 vs 453 and 471). The low proportion of intracellular proteins detected in our dataset demonstrates the advantage BONCAT provides by enabling FBS supplementation, which is essential for cell viability. It must be noted, however, the undefined nature and batch variability of FBS can make

the study less generalizable, and that FBS contamination is still present, in up to 138/742 of proteins (18.6%). In addition, there could be a potential bias for Met-rich peptide sequences. Finally, although AHA pulse did not affect expression of adipocyte and thermogenic genes tested, its full biological effect is unknown and may include protein misfolding and deficiency in single-carbon metabolism. The remaining 20% of proteins not predicted or annotated to be secreted include intracellular proteins such as tubulin and actin. However, many are non-classically secreted proteins such as FABP4/5 (*Hotamisligil and Bernlohr, 2015*). Additionally, many proteins (such as GAPDH) that were initially thought to be leakage products turn out to be secreted in a regulated fashion (*Takenouchi et al., 2015*), and our dataset includes other novel factors we have independently confirmed as bona fide secreted factors. Our findings can therefore greatly expand the scope of the adipose secretome that includes non-classical modes of secretion, but each factor will need to be carefully studied to validate whether its presence in the media is from a regulated secretion process or from non-specific leakage.

We have also demonstrated that BONCAT can label nascent serum proteins in vivo. This technique can be further utilized to study the nascent serum proteome in response to stimuli difficult to model in vitro, such as obesity and changes in ambient temperature. Novel technologies for in vivo profiling of cell type-specific proteomes using non-canonical amino acids (*Alvarez-Castelao et al., 2017*) or proximity labeling (*Branon et al., 2018*; *Wei et al., 2021*) are on the horizon. This study provides a baseline profile of the nascent serum proteome in lean adult mice from which cell type enrichment analyses can be performed, and bioinformatic pipelines implemented here to compare tissue expression would be useful in cross-referencing with transcriptomic datasets.

AAV-mediated LRG1 overexpression in the adipose tissues led to a constellation of metabolic effects in both B6 and *db/db* mice, resulting in improvement of glucose/insulin tolerance as well as a reduction of obesity-associated inflammation. While endogenous LRG1 levels increase with obesity, further induction of LRG1 expression via AAV transduction protected mice from obesity-related complications. These findings suggest that while LRG1 plays a protective role in obesity, the degree of LRG1 induction is not sufficient, and that AAV-mediated overexpression could be augmenting this physiologic response. LRG1 overexpression had a more dramatic effect in the *db/db* model, likely due to the severity of inflammation and metabolic derangements. In both mouse models, LRG1 overexpression led to an improvement in fasting glucose levels, which also contributed to differences in GTT and ITT profiles. We believe it is the improvement in insulin sensitivity that drives both findings, given increased IRS-1 protein levels in eWAT and lack of difference in hepatic gluconeogenic gene expression or pyruvate tolerance.

Adipose tissue expansion in mice is characterized by periods of adipocyte turnover, during which there is a sharp increase in adipocyte death and monocyte/macrophage accumulation (*Rosen and Spiegelman, 2014*; *Weisberg et al., 2003*). In B6 eWAT, adipocyte death and CLS formation peak around 12–16 weeks on HFD (*Strissel et al., 2007*), which corresponds to when a parallel surge in fasting blood glucose levels is observed. In *db/db* mice, the remodeling seems to be initiated earlier, as CLS in WAT emerge at 7 weeks of age, and chemokine/cytokine expression and circulating levels are more highly induced at week 7 than week 10. Periods when LRG1's insulin-sensitizing and anti-inflammatory effects are particularly effective coincide with these times of adipocyte death and turnover. In *db/db* mice, we also observed accelerated WAT expansion with LRG1 overexpression. Adipose tissue hypertrophy likely drives this process, given larger adipocyte cross-sectional areas and LRG1's lack of adipogenic effect. As inflammation is a key mechanistic link between obesity and insulin resistance (*Saltiel and Olefsky, 2017*), these findings strongly suggest LRG1 promotes insulin sensitization via modulation of inflammation and cell death during adipose tissue remodeling.

Recently, whole-body LRG1 loss of function has been reported to reduce obesity and improve insulin sensitivity by reduction of hepatosteatosis (*He et al., 2021*). In the LRG1-KO mice we generated, we have not observed any effect on body weight or glucose/insulin tolerance profiles. This discrepancy is potentially due to differences in how the knockout lines were generated (gene cassette vs. CRISPR-Cas9). The work presented here sheds light on the biology of adipose-derived LRG1 with AAV overexpression, which has not been studied before. Our work also expands our understanding of LRG1 function in its native tissue environment, the adipose tissue, where it reduces inflammation and cell death.

Our studies build on our understanding of the LRG1-Cyt *c* interaction that has thus far been studied in cell culture models (*Codina et al., 2010*; *Jemmerson et al., 2021*). We propose LRG1 functions

as a buffer against deleterious effects of extracellular Cyt *c* released from dying/dead cells. During obesity-driven adipose tissue remodeling, increase in adipocyte death contributes to increased circulating Cyt *c*, which could be an important pro-inflammatory signal triggering monocyte/macrophage recruitment. The pro-inflammatory setting of obesity also induces LRG1 in other adipocytes, and this increase in circulating LRG1 modulates inflammation by directly binding Cyt *c*. Of note, both TLR4 and LRG1 contain LRR domains, so the Cyt *c*-neutralizing effect by LRG1 could involve steric interference as the two proteins compete for binding Cyt *c*. It is interesting to note that LRG1 overexpression led to reduction in circulating Cyt *c* levels. As our data suggest LRG1-Cyt *c* complex formation inhibits renal excretion of Cyt *c*, the reduction of serum Cyt *c* in the context of LRG1 overexpression is most consistent with decreased Cyt *c* release from tissues. Together with our data showing reduction of Cyt *c* in the ISF of eWAT and liver from LRG1-overexpressing mice, LRG1's immunomodulatory role in these metabolic tissues may contribute to decreased Cyt *c* release by reducing tissue damage. Many questions remain on how LRG1 and Cyt *c* regulate inflammation and cell death. Whether LRG1-Cyt *c* interaction affects tissue-resident macrophages or also impacts immune cell recruitment remain to be studied. Mechanistically, as extracellular Cyt *c* is pro-apoptotic, it is possible that LRG1-Cyt *c* binding directly prevents apoptosis either by steric hindrance or by simultaneous binding of LRG1 to Cyt *c* and TGF-β1 (*Jemmerson, 2021*). The scope of LRG1's metabolic function beyond its interaction with Cyt *c* will also continue to be explored in our future work.

While the current study focused on LRG1's effect on adipose tissue, it is important to emphasize that LRG1-mediated anti-inflammatory effects were systemic, as observed by the decrease in inflammatory foci in LRG1-overexpressing *db/db* liver. As hepatocyte apoptosis contributes to progression of NASH (non-alcoholic steatohepatitis) in NAFLD (*Feldstein et al., 2003*), further investigation may uncover a hepato-protective role for LRG1 in NAFLD. Multiple biomarker studies in humans have shown that increased LRG1 is associated with not only obesity (*Pek et al., 2018*) but also conditions involving cell death and inflammation, such as appendicitis (*Kentsis et al., 2010*), solid tumors (*Belczacka et al., 2019*) and autoimmune diseases (*Naka and Fujimoto, 2018*). Our study suggests that induction of LRG1 in these settings could be a compensatory mechanism to modulate inflammation.

Enhanced renal clearance of Cyt *c* in LRG1-KO mice provides further evidence that LRG1 complexes with Cyt *c* in circulation and highlights the need to measure LRG1 levels in future studies utilizing extracellular Cyt *c* as a biomarker. Comparable serum Cyt *c* half-lives between WT and LRG1-KO suggest LRG1 loss of function does not affect kidney function. Divergence in Cyt *c* clearance profiles occurs towards the plateau, and we believe this difference represents the portion of Cyt *c* that stays bound to LRG1 in WT animals. Inflammation in obesity, especially in B6 DIO mice, is chronic and relatively low grade. Given our observation that HFD-fed LRG1-KO mice have reduced Cyt *c* levels in circulation, enhanced clearance of Cyt *c* in LRG1-KO mice could serve to counteract Cyt *c* release and protect from its deleterious effects. It is possible therefore that the effect of LRG1 loss of function may be more pronounced in conditions involving acute or high-grade inflammation. It is interesting to note that in streptozotocin (STZ)-induced diabetic mouse models, LRG1 loss of function had a protective role in development of diabetic nephropathy (*Hong et al., 2019*). In addition to the reported LRG1-mediated aberrant glomerular angiogenesis as a mechanism for this finding, glomerular deposition of LRG1-Cyt *c* and subsequent immune response could be another potential pathogenic mechanism.

In summary, we identified LRG1 as a novel adipokine and discovered its metabolic role as an insulin sensitizer and suppressor of inflammation. LRG1's in vivo function is associated with Cyt *c* binding, and this interaction sheds light on previously unappreciated molecular players of adipocyte-macrophage interaction during adipose tissue expansion and remodeling. LRG1 or targeting extracellular Cyt *c* could be an attractive therapeutic approach for treatment of not only obesity but a variety of inflammatory conditions.

## Methods
### Animals
C57BL/6 J mice were purchased from Jackson Laboratories. C57BLKS-*Lepr^{db}* homozygote males and C57BLKS-*Dock7^m* homozygote males were purchased from Jackson Laboratories. LRG1-KO (*Lrg1^{-/-}*) line was generated in C57BL/6 J background by the CRISPR and Gene Editing Center at the Rockefeller University and backcrossed with C57BL/6 J mice for at least five generations to minimize

off-target effects. Cohorts of LRG1-KO and WT littermates for characterization studies were obtained by intercrossing male and female heterozygotes. Animals were maintained at the Rockefeller University Comparative Biosciences Center, housed at 23 °C and maintained at 12 hr light:dark cycles. The animals were group-housed with ad libitum access to food and water except during metabolic characterization studies. C57BL/6 J mice were fed standard chow diet (LabDiet 5053), and where specified, started on 60% high fat diet (HFD, Research Diets D12492) feeding at 6 or 8 weeks of age. C57BLKS-*Lepr*^db and C57BLKS-*Dock7*^m were fed standard chow diet. For virus injections, each cage containing 4 mice was randomly assigned to eGFP or LRG1 group. Injections were performed in designated ABSL-2 housing rooms and the transduced mice were quarantined for 72 hr post-injection before transferred to regular housing rooms. Experiments involving adenoviral and AAV8 vectors were performed in accordance with the institutional ABSL-2 guidelines. Except in the case of mortality, no mice were excluded from study analysis. AAV experiments consisted of 2 B6 cohorts and 3 *db/db* cohorts. All animal studies were performed in accordance with the institutional guidelines of the Rockefeller University Institutional Animal Care and Use Committee (IACUC).

## Metabolic characterization of mice

For studies involving diet-induced obesity, mouse body weights were monitored once a week, providing fresh 60% high fat diet (Research Diets) at least once a week. For fasting blood glucose measurements, mice were single housed in the morning in cages with fresh bedding and access to water but without food. Mice were kept in a procedure room free of noise or vibration throughout the experiment. After 6 hr, blood was collected from the tail vein, and glucose levels were measured using a glucose meter. For plasma insulin ELISA, blood was also collected in EDTA-coated capillary tubes, which were centrifuged at 2000×g for 15 min at 4 °C to collect plasma. Insulin ELISA was performed using Ultra Sensitive Mouse Insulin ELISA Kit (Crystal Chem). Glucose (GTT), pyruvate (PTT) and insulin tolerance tests (ITT) were performed following a similar 6 hr fasting procedure and started by administering at time 0 indicated doses of glucose or Novolin R human insulin (Novo Nordisk) by intraperitoneal injection or sodium pyruvate by oral gavage. Following injection/gavage, blood glucose measurements were taken from the tail vein at indicated timepoints. During an ITT procedure, mice with glucose measurements below 20 mg/dL or showing signs of hypoglycemia were rescued by 1 g/kg glucose IP injection and excluded from the study. For glucose-stimulated insulin secretion (GSIS) assay, mice were fasted for 6 hr, followed by intraperitoneal injection of glucose and subsequent blood glucose measurement and blood collection at indicated intervals. Plasma insulin levels were measured as described above. Plasma non-esterified fatty acid (NEFA) concentrations were measured using the HR Series NEFA-HR(2) assay (Wako/Fujifilm).

## Indirect calorimetry

Phenomaster automated home cage phenotyping system (TSE Systems, Bad Homburg, Germany) was used for the study. Mice were housed individually in controlled environmental chambers maintained at 22 °C, 12 hr light:dark cycles, and 40% humidity. Mice were given ad libitum access to food and water, and bedding was changed every 5 days. For $O_2$ and $CO_2$ measurements, gas was sampled at 15-min intervals with a settle time of 3 min, at 0.25 L/min sample flow rate. Raw values were analyzed using CalR (*Mina et al., 2018*). Daily food intake was measured directly by weighing remaining food pellets in each cage.

## Generation of LRG1-KO mice

CRISPR guide RNAs were designed using CRISPOR.org (*Concordet and Haeussler, 2018*) and were used as two-part synthetic crRNA and tracrRNA (Alt-RTM CRISPR guide RNA, Integrated DNA Technologies, Inc). Cas9 protein, crRNA, and tracrRNA were assembled to ctRNP using protocols described previously (*Shola et al., 2021*). Two crRNAs were assembled to ctRNPs and electroporated to one-cell-stage mouse embryos to assess their efficiency in generating indels on the Exon 2 of *Lrg1* gene. To prepare for the microinjection mix, crRNA-B which binds to genomic target sequence " AATCTCGGTGGGACCATGGCAGG" was selected for its high on-target efficiency and low off-target potential. The final injection mix was made of 0.6 µM of guide RNA (crRNA +tracrRNA) and 0.3 µM of Cas9 protein according to protocols described previously (*Shola et al., 2021*). The injection mix was

then delivered to 0.5 days of fertilized C57BL/6 J mouse embryos using well-established pronuclear injection and surgical protocols (*Shola et al., 2021*).

## Cells

Mouse primary stromal vascular fraction (SVF) cells were obtained from adipose tissues of 6- to 8-week-old male mice by collagenase digestion and plated on collagen I-coated dishes. SVF cells from epididymal white adipose tissue (eWAT) were grown in ITS media containing 1.5:1 mixture of low-glucose DMEM:MCDB201 supplemented with 2% FBS (Gemini), 1% ITS premix (Corning), 0.1 mM L-ascorbic acid 2-phosphate (Sigma), 10 ng/mL bFGF (Thermo), 0.5% penicillin/streptomycin (P/S, Gibco), and 0.2% primocin (InvivoGen). SVF cells from inguinal white adipose tissue (iWAT) and inter-scapular brown adipose tissue (BAT) were grown in DMEM/F-12 GlutaMAX medium (Gibco) containing 10% FBS and 1% P/S. Once grown to confluence, differentiation and maintenance of primary adipo-cytes across cell types were done using DMEM/F-12 GlutaMAX medium containing 10% FBS and 1% P/S. eWAT and iWAT SVF cells were induced to differentiate with an adipogenic cocktail (0.5 mM IBMX, 1 μM dexamethasone, 1 μM rosiglitazone, and 850 nM insulin) for the first 2 days, followed by 2 days of 1 μM rosiglitazone and 850 nM insulin, after which the cells were maintained in 850 nM insulin for additional 2–4 days. SVF cells from BAT were differentiated as above but with 17 nM insulin. Experiments with primary adipocytes were performed between days 6 and 8 of differentiation. All cultured primary adipocytes were checked for lipid accumulation under a phase-contrast microscope before studies. Primary SVF and adipocytes were maintained at 37 °C with 10% $CO_2$.

Bone marrow-derived macrophages (BMDMs) were obtained from 8- to 10-week-old males. Femurs and tibias were dissected, cleaned, and sterilized with ethanol before flushed of bone marrow cells, which were plated onto petri dishes. Bone marrow cells were differentiated in RPMI-1640 medium (Gibco) supplemented with 20% heat-inactivated FBS (Sigma), 1% P/S, and 100 ng/mL M-CSF (Biolegend) for 6–7 days, changing media every 2–3 days. Differentiated BMDMs were washed, trypsinized, and plated onto TC-treated culture plates for overnight before studies were performed. Experiments with BMDMs were performed between days 6 and 7 of differentiation, in X-VIVO 15 medium (Lonza). BMDMs were differentiated and maintained at 37 °C with 5% $CO_2$.

HEK293A cells were purchased from Invitrogen and grown using 4.5 g/L glucose DMEM (Gibco) supplemented with 10% FBS and 1% P/S at 37 °C with 5% $CO_2$. Cells with passage number under 20 were used for adenovirus production. Cells were validated to be mycoplasma free.

## Oil Red O

Cells were washed with PBS and fixed with 10% formalin in PBS. Cells were washed with distilled water and 100% propylene glycol. 0.5% Oil Red O staining solution in propylene glycol was added and cells and incubated overnight at room temperature. Following staining, cells were washed with 85% propylene glycol solution followed by washes with distilled water. Stained cells were stored in 50% of glycerol.

## Proteomic analysis using BONCAT

### Conditioned medium generation

On day 6 of differentiation, primary adipocytes on collagen-coated 6-well plates were washed twice with warm PBS and pulsed with 1 mL/well of Met-free DMEM containing 10% dialyzed FBS, 1% P/S, 17 nM or 850 nM insulin, and either 0.1 mM AHA or 0.1 mM Met. Following 24 hr incubation at 37 °C with 10% $CO_2$, conditioned media (CM) from six wells (1 plate) were collected and pooled, filtered through a 0.22 μm PES membrane syringe filter unit, and supplemented with ½ tabs of EDTA-free cOmplete mini protease inhibitor cocktail (Roche) and PhosSTOP (Roche). CM was concentrated using a 3 kDa centrifugal filter unit (Millipore).

### AHA administration and serum collection

Mice were injected with 0.1 g/kg/day AHA or PBS IP for 2 consecutive days and sacrificed 24 hr following the second injection. Following decapitation, truncal blood was collected, allowed to clot for 15 min at room temperature, and centrifuged at 2000×g for 15 min at 4 °C to collect serum.

## In-gel fluorescence analysis

Concentrated CM or serum was dialyzed with phosphate-buffered RIPA (10 mM phosphate buffer pH 7.2, 1% Triton X-100, 0.1% Na deoxycholate, 0.1% SDS, 140 mM NaCl) supplemented with EDTA-free cOmplete mini protease inhibitor cocktail (Roche) and PhosSTOP (Roche) using a 3 kDa centrifugal filter unit (Millipore) and protein concentration was determined using Pierce BCA Protein Assay Kit (Thermo Scientific) using a dilution series of bovine serum albumin as protein standards. Copper(I)-catalyzed azide-alkyne cycloaddition reaction with TAMRA-alkyne (Invitrogen) was performed by mixing 200 μg of CM proteins with 0.1 mM TAMRA-alkyne, 1 mM TCEP, 0.1 mM TBTA, and 1 mM of $CuSO_4$ in phosphate-buffered RIPA and rotated end-over-end for 1 hr at room temperature under protection from light. Following methanol/chloroform precipitation, the dried protein pellet was dissolved in Laemmli loading buffer. Following polyacrylamide gel electrophoresis, the gel was briefly washed with distilled $H_2O$ and imaged with a Typhoon 5400 imager (GE Healthcare) using 532 nm excitation and a 580 nm detection filter.

## Enrichment of labeled proteins

Azide-labeled nascent protein in the concentrated CM was enriched using Click-iT Protein Enrichment Kit (Invitrogen). Serum was diluted 1:1 with the lysis buffer provided with the kit and subjected to enrichment. Enrichment and resin wash was performed following the protocol from *Eichelbaum and Krijgsveld, 2014*.

## On-bead digestion

Extensively washed beads were incubated with Lys-C endopeptidase (Wako) in 4 M urea and 0.14 M $NH_4HCO_3$ by shaking at 1400 rpm for 6 hr at room temperature. The resin mixture was further digested by adding trypsin (Promega) in 2 M urea and 0.14 M $NH_4HCO_3$ and incubated by shaking at 1400 rpm overnight at room temperature. The following day, the digestion reaction was quenched by adding trifluoroacetic acid.

## LC-MS/MS

Tryptic peptides were desalted (*Rappsilber et al., 2007*) and separated by reverse phase nano-LC-MS/MS (column: 12 cm/75um C18 built-in-emitter column, Nikkyo Technos Co., Ltd. Japan, EasyLC 1200, Thermo Scientific) using a 70-min analytical gradient, increasing from 2% B/98% A to 38%B/62% A (A: 0.1% formic acid, B: 80% Acetonitrile/0.1% formic acid) at 300 nL/min. The mass spectrometer (Fusion Lumos, Thermo Scientific) was operated in high/high mode (120,000 and 30,000 for MS1 and MS2, respectively). Auto Gain Control was set at 50,000 for MS2. MS1 scan range was set to m/z 375–1500 and m/z 110 was set as lowest recorded mass in MS2. One-point lock mass calibration was used. All data were quantified and searched against a Uniprot mouse database using MaxQuant (v. v. 1.6.0.13) (*Cox et al., 2014*). Oxidation of methionine and protein N-terminal acetylation were allowed as variable modifications, cysteine carbamidomethyl was set as a fixed modification, and two missed cleavages were allowed. The 'match between runs' option was enabled, and false discovery rates for proteins and peptides were set to 1%. Protein abundances measured using label free quantitation (*Tyanova et al., 2016*).

## Proteomic data analysis

Proteomic datasets were analyzed using Perseus v1.6.14.0 (*Tyanova et al., 2016*). Of the detected proteins, those flagged as reverse, only identified by site, and potential contaminants were excluded from the analysis. For quantitative analysis, LFQ or iBAQ intensities were employed as indicated; LFQ intensities were used for comparisons across samples, while iBAQ intensities were used to compare abundances across different proteins. Imputation of undetected data points for $Log_2$(LFQ) intensities was performed by assigning values from a normal distribution of 0.3 width and 1.8 down shift. Principal component analysis (PCA) was performed with imputed $Log_2$(LFQ) intensities. Scatterplot representation of $Log_2$(LFQ) intensities was generated without imputation. Differentially secreted proteins were identified with ANOVA using permutation-based FDR, with FDR set at 0.01 and number of randomizations at 250.

## GO cell component analysis

Gene symbols from detected proteins were submitted to Retrieve/ID mapping tool on the UniProt website (https://uniprot.org). List of genes that are annotated with the following gene ontology cell component terms were obtained: extracellular region (5576), extracellular space (5615), extracellular matrix (31012), plasma membrane (5886), cytosol (5829), nucleus (5634), mitochondrion (5739), endoplasmic reticulum (5783), and Golgi apparatus (5794).

## Secretion prediction analysis

UniProt accession IDs of the detected proteins were submitted to Retrieve/ID mapping tool on the UniProt website (https://uniprot.org) to obtain the FASTA sequences, which were used as inputs for various secretion prediction algorithms using the web-based query system. We defined classically secreted proteins as having SignalP5.0 score >0.5 and 0 or 1 predicted transmembrane domains by TMHMM2.0. Subcellular localization prediction analysis was performed using DeepLoc1.0 and searched for proteins whose predicted location is extracellular. PredGPI specificity score >99% was used as the threshold to determine if a protein is expected to be GPI-anchored. P roteins with SecretomeP2.0 score >0.6 and SignalP5.0 score ≤0.5 were considered non-classically secreted.

## Cluster analysis and functional annotation

Hierarchical clustering was performed on z-score-transformed $Log_2$(LFQ) values using the complete-linkage method and split into 4 clusters by dendrogram. DAVID v6.8 was used to generate functional annotation of clusters (*Huang et al., 2009*). List of genes encoding the proteins of each of the 4 clusters were compared to a background gene list of total detected proteins in the proteomic dataset. Per cluster, top 4 overrepresented pathways in the gene ontology biological process terms were reported.

## Adipose tissue enrichment analysis

To identify candidate genes enriched within brown and white adipose relative other tissue types as well to each other we used the BioGPS datasets (*Su et al., 2004*). The Mouse GNF1M Gene Atlas datasets (GSE1133) were downloaded from BioGPS portal (*Su et al., 2004*) and imported into Limma Bioconductor package (*Ritchie et al., 2015*) for Log2 transformation and differential expression analysis. All pair-wise comparisons for both brown adipose and white adipose tissues against all other tissue types were performed using limma as well as the direct comparison between brown and white adipose tissues. Genes with a Log2 fold change greater than 4 and a Benjamini-Hochberg-corrected FDR of 0.05 within pair-wise comparisons were considered significantly enriched. Genes were further scored by the total number of pair-wise comparisons where genes were found to be enriched in both adipose tissues, brown adipose tissue or white adipose tissue compared to other tissues in the tissue atlas.

## RNA isolation, cDNA synthesis, and RT-qPCR

Total RNA was extracted from cultured cells using RLT buffer (Qiagen) and from tissues using TRIzol (Invitrogen) and purified using RNeasy Mini Kit (Qiagen). cDNA was synthesized from 1 µg of RNA using the High-Capacity cDNA Reverse Transcription Kit (Applied Biosciences). Power SYBR Green (Life Technologies) was used for RT-qPCR reactions performed with QuantStudio 6 Flex Real-Time PCR System (Thermo Scientific) in a 384-well format. Relative fold changes of mRNA levels were calculated using the ΔΔCT method with 18 S rRNA as loading control. qPCR primers are provided in *Supplementary file 3*.

## Adipose tissue fractionation

Adipose tissues from 8-week-old C57BL/6 J WT male mice were dissected and minced. eWAT and iWAT were digested in a buffer containing 10 mg/mL collagenase D (Roche), 2.4 mg/mL Dispase II (Roche), and 10 mM $CaCl_2$ in PBS. For BAT, 2 x BAT digestion buffer containing 125 mM NaCl, 5 mM KCl, 1.3 mM $CaCl_2$, 5 mM glucose, 1% P/S, and 4% BSA was prepared, which was diluted 1:1 with PBS and used to dissolve collagenase B (Roche) at a final concentration of 1.5 mg/mL. Following collagenase digestion of the tissues in a 37 °C water bath, the mature adipocyte fraction was separated from the SVF pellet by centrifugation at 500×g for 10 min at 4 °C. The two fractions were transferred to two

separate tubes, washed with DMEM/F-12 GlutaMAX containing 10% FBS and 1% P/S, and vortexed in TRIzol for RNA extraction.

## RNA-Sequencing and immune cell deconvolution

Extracted RNA samples were analyzed for RNA integrity number (RIN) using the Bioanalyzer (Agilent) and sequenced using Illumina NovaSeq at the Rockefeller University Genomics Resource Center. Reads were trimmed with Cutadapt, aligned to mm10 reference genome using STAR, and quantified using featureCounts. Differential gene expression analysis was performed using DESeq2 (*Love et al., 2014*). Pathway analysis was performed using clusterProfiler (*Yu et al., 2012*). Deconvolution analysis was performed with CIBERSORTx (*Newman et al., 2019*) using ImmuCC signature matrix (*Chen et al., 2017*).

## Tissue expression pattern analysis

Mouse tissue mRNA sequencing data from ENCODE was downloaded from GSE36026. Reads were mapped and quantified as above and gene expression was normalized using DESeq2. 177 genes encoding 180 proteins detected in mouse nascent serum were selected for t-SNE analysis. Briefly, average normalized expression in a tissue was divided by summed expression across tissues. Tissue with the highest relative expression was designated as the highest expressing tissue for a gene. t-SNE analysis was performed on relative expression values with R package Rtsne version 0.16 (*Krijthe, 2015*) using a perplexity of 30 and maximum iteration of 1000.

## Interstitial fluid collection

Mice were sacrificed, then carefully perfused with approximately 20 mL dPBS. Whole tissues were carefully excised with minimal tissue disturbance and laid flat onto a 70 μm nylon mesh over a tube containing 2 μL of 10 x protease inhibitor solution (Roche). Tubes were sealed and centrifugated at 300 g at 4 °C. Following collection, aqueous infranatant was collected for downstream processing.

## Immunoblot

Upon collection, conditioned medium (CM) was filtered using a 0.22 μm PES membrane syringe filter unit to remove cell debris and concentrated using a 3 kDa centrifugal filter units (Millipore). Tissue protein lysates were collected by homogenizing flash-frozen tissue in RIPA buffer (10 mM Tris pH 8.0, 1 mM EDTA, 1% Triton X-100, 0.1% Na deoxycholate, 0.1% SDS, and 140 mM NaCl) containing cOmplete mini protease inhibitor cocktail (Roche) and PhosSTOP (Roche). Protein concentration was determined using Pierce BCA Protein Assay Kit (Thermo Scientific) using a dilution series of bovine serum albumin as protein standards. Mouse serum samples were loaded at equal volume. Pre-cast polyacrylamide gels were used for electrophoresis, after which protein was transferred to PVDF membrane using standard techniques. Immunoblots were incubated with indicated primary antibodies and developed using Western Lightning Plus-ECL (PerkinElmer) and imaged on an autoradiographic film or using a Bio-Rad Gel Doc system.

## Generation of LRG1-depleted FBS

Affi-Gel10 (Bio-Rad) was washed with cold distilled water and resuspended in binding buffer (0.2 M NaHCO3 and 0.5 M NaCl). Equine Cyt *c* was dissolved in binding buffer at 40 mg/mL concentration and conjugated with Affi-Gel10 for 1 hr at room temperature. Reaction was quenched with buffer containing 0.5 M ethanolamine and 0.5 M NaCl pH 8.3 for 1 hr. Conjugated beads were washed with copious amounts of PBS and elution buffer (0.1 M acetate and 0.5 M NaCl pH 4.0). For depletion of LRG1 from FBS, 2 mL of 50% Cyt *c*-agarose bead suspension was added to 50 mL of FBS and incubated overnight at 4 °C and repeated once more with fresh conjugated beads. FBS was sterile-filtered before use. Specific binding of LRG1 to Cyt *c*-agarose beads was confirmed by incubating the beads used for depletion with elution buffer for 10 min at room temperature and subjecting the supernatant to SDS-PAGE electrophoresis followed by Coomassie stain or western blot against LRG1.

## Co-immunoprecipitation

Serum was diluted 1:1 with PBS containing 0.02% Tween-20 (PBS-T) and incubated with α-FLAG M2 beads (Sigma) overnight at 4 °C. Following wash with PBS-T, bound proteins were eluted by

heating the beads at 70 °C in Laemmli buffer containing 50 mM glycine buffer pH 2.8 and 9% (v/v) β-mercaptoethanol.

## Adenovirus production and purification

Adenoviral vectors were created using the AdEasy system (*Luo et al., 2007*). C-terminally 3xFLAG-tagged murine LRG1 was cloned into pAdTrack-CMV (AddGene) linearized with XhoI (NEB) and HindIII (NEB) using In-Fusion HD Cloning Kit (Takara). pAdTrack-CMV and pAdTrack-CMV-LRG1-FL plasmids were linearized with PmeI (NEB) and recombined into pAdEasy-1 vector via electrophoretic transformation of recombination-competent BJ5183-AD-1 cells (Agilent) with the linearized product and selection for kanamycin-resistant clones. Plasmids from validated clones were transformed into recombination-deficient XL-10 Gold ultracompetent cells (Agilent), which were used to generate pAd-eGFP and pAd-LRG1-FL plasmids and purified using Plasmid Maxi Kit (Qiagen).

Crude adenovirus was produced by transfecting PacI (NEB)-linearized pAd vectors into HEK293A cells (Invitrogen), which were incubated at 37 °C with 5% $CO_2$ for 10–14 days with media supplementation every 3–5 days until most cells showed cytopathic effect/detachment. Both cells and the culture medium were collected, lysed by 3 cycles of freeze-thaw between dry ice-ethanol and room-temperature water baths, and centrifuged at 3500×g for 15 min at 4 °C to obtain the supernatant crude virus. Round 1 amplification product was obtained by transducing HEK293A cells with the crude virus and repeating the above collection, lysis, and centrifugation steps.

To obtain round 2 amplification product, twelve 15 cm plates of HEK293A cells were transduced with round 1 adenovirus and incubated at 37 °C with 5% $CO_2$ until most cells demonstrated cytopathic effect. As with previous rounds, cells and media were collected, lysed by freeze-thaw cycles, and centrifuged to obtain the supernatant. The supernatant was treated with benzonase, and adenoviral particles were purified from the crude mixture using the Vivapure AdenoPACK 100 kit (Sartorius). Purified virus was dialyzed with buffer containing 20 mM Tris pH 8, 25 mM NaCl, and 2.5% (w/v) glycerol and concentrated using a 100 kDa centrifugal filter unit provided with the kit. Titer of the adenovirus was determined using Adeno-X Rapid Titer Kit (Takara).

## AAV8 vector preparation

C-terminally 3xFLAG-tagged LRG1 was cloned into pENN.AAV.CB7.CI.eGFP.WPRE.rBG (Addgene) linearized with EcoRI (NEB) and BglII (NEB) using In-Fusion HD Cloning Kit (Takara). The original eGFP-expressing and cloned LRG1-FL plasmids were transformed into Stable Competent *E. coli* (NEB), purified using Plasmid Maxi Kit (Qiagen), and shipped to Penn Vector Core (PA, USA) for AAV8 production.

## In vivo adenovirus/AAV8 transduction

In vivo adenoviral transduction studies were performed using purified adenovirus from second round of amplification. Adenovirus was injected at a dose of $10^{10}$ pfu/mouse. AAV8 was injected at $10^{11}$ GC/mouse. The mice were briefly anesthetized with isoflurane for virus injection via the retroorbital route. Following injection, the mice were quarantined in an ABSL-2 housing room for 72 hr before transferred back to regular housing conditions.

## Histology

Dissected tissues were fixed in 10% neutral buffered formalin for 3 days at room temperature and transferred to 70% ethanol. Paraffin embedding, sectioning, and H&E staining was performed by the Memorial Sloan Kettering Cancer Center Laboratory of Comparative Pathology. H&E-stained sections were imaged using a wide-field fluorescence/brightfield/DIC microscope (Zeiss) at the Rockefeller University Bioimaging Resource Center. Crown-like structures (CLS) were identified as any adipocyte in a field of view with cellular infiltrates indicated by nuclear staining surrounding a majority of adipocyte perimeter. Objectives were used as indicated and chosen based on CLS enumerability (<30 CLS per field). 5 fields from 3 animals were quantified per group. Adipocyte area was measured from a minimum of 289 adipocytes from 2 to 4 animals per group using Cellpose 2.0 for image segmentation (*Stringer et al., 2021*).

## Liver Triglyceride Quantification

Per 100 mg of flash-frozen liver, 1 mL of 5% NP-40 lysis buffer was added to generate a homogenate. The homogenized tissue samples were then subject to two cycles of heating at 80–100°C for 5 minutes followed by cooling at room temperature. The resulting mixture was centrifuged, and supernatant was collected. Triglyceride concentration of the supernatant was measured using Triglyceride Quantification Colorimetric/Fluorometric Kit (Sigma, MAK266) following manufacturer instructions.

## Multiplex cytokine panel

Mouse serum samples were diluted 1:1 with PBS, snap frozen using liquid $N_2$ and shipped to Eve Technologies (Alberta, Canada) on dry ice. The Mouse Cytokine Array / Chemokine Array 31-Plex (MD31) panel was used to quantify the levels of cytokines and chemokines.

## Cytochrome *c* clearance assay

12-week-old chow-fed LRG1-KO and WT littermate males were injected retro-orbitally with 40 mg/kg equine cytochrome *c* (Cyt *c*) in PBS. Blood was collected from the tail vein immediately prior to injection (0 min) and 5 min, 15 min, 30 min, 1 h, 2 h, 4 h, and 6 h post-injection into EDTA-coated capillary tubes and kept in ice until further processing. During blood collection, each mouse was placed on a metal grating above a clean plastic wrap to allow collection of excreted urine, if any. Plasma was isolated via centrifugation at 2000×g for 15 min at 4 °C. Immunoblot against Cyt *c* was performed with WT and KO plasma samples run pairwise to enable relative quantification of Cyt *c* signal. Quantification was performed using ImageJ (*Schneider et al., 2012*).

## Statistical analysis

Unless otherwise noted, data are presented as mean ± SEM, with *n* number specified in the figure legends. Statistical analyses were performed with GraphPad Prism 9. Binary comparisons were performed with Welch's *t*-test to account for possible difference in variance. Statistical analysis of data involving 3 or more conditions (levels) of a single variable was performed using one-way ANOVA followed by post hoc tests using the Dunnett method to compare every mean with a control mean or the Tukey method to compare every mean with every other mean. Data measured across multiple time points as in GTT, ITT, and PTT were analyzed with repeated measures two-way ANOVA, reporting group factor p-values. Analysis of data from a two-factor experimental setup was performed with two-way ANOVA or two-way mixed effects ANOVA in the case of an uneven *n* number, reporting group factor p-values. For post hoc tests, the Tukey method was used when comparing every mean with every other mean and the Šídák method was employed when a selected set of means were compared.

## Materials availability

The mass spectrometry proteomics data have been deposited to the ProteomeXchange Consortium via the PRIDE partner repository with the dataset identifier PXD035318. RNA-Seq data have been deposited to GEO with the identifier GSE208219. Reagents including unique biological materials are available from the corresponding author upon request.

## Code availability

Publicly available codes are available in the relevant references.

## Acknowledgements

We thank Jeffrey M Friedman, James C Lo, Howard C Hang, Jingyi Chi, and Kaja Plucińska for feedback and discussions. We also thank Chingwen Yang of the CRISPR and Genome Editing Center at the Rockefeller University for generating the LRG1-KO line. We are grateful to Tao Tong and Alison North of the Bio-Imaging Resource Center at the Rockefeller University for assistance with microscopy. CHJC and SKS were supported by a Medical Scientist Training Program grant from the National Institute of General Medical Sciences of the National Institutes of Health under award number T32GM007739 to the Weill Cornell/Rockefeller/Sloan Kettering Tri-Institutional MD-PhD Program. SZ was supported by the Sarnoff Cardiovascular Research Fellowship Program. PC was supported by the American Diabetes Association Pathway to Stop Diabetes Grant 1–17-ACE-17 and by NIH grant RC2 DK129961. Data

was generated by the Proteomics Resource Center at The Rockefeller University (RRID:SCR_017797) using instrumentation funded by the Sohn Conferences Foundation and the Leona M and Harry B Helmsley Charitable Trust. Bio-Render was used to make the schematic diagrams. The content of this study is solely the responsibility of the authors and does not necessarily represent the official views of the National Institutes of Health.

## Additional information

### Funding

| Funder | Grant reference number | Author |
| --- | --- | --- |
| American Diabetes Association | 1-17-ACE-17 | Paul Cohen |
| National Institutes of Health | RC2 DK129961 | Paul Cohen |
| National Institute of General Medical Sciences | T32GM007739 | Chan Hee J Choi Sarah K Szwed |
| Sarnoff Cardiovascular Research Foundation | | Samir Zaman |

The funders had no role in study design, data collection and interpretation, or the decision to submit the work for publication.

### Author contributions

Chan Hee J Choi, Conceptualization, Data curation, Formal analysis, Validation, Investigation, Visualization, Methodology, Writing – original draft, Writing – review and editing; William Barr, Samir Zaman, Data curation, Formal analysis, Validation, Investigation, Visualization, Methodology, Writing – review and editing; Corey Model, Francois Marchildon, Audrey Crane, Investigation; Annsea Park, Software, Visualization, Methodology; Mascha Koenen, Zeran Lin, Sarah K Szwed, Investigation, Methodology; Thomas S Carroll, Software, Methodology; Henrik Molina, Resources, Software, Methodology; Paul Cohen, Conceptualization, Resources, Supervision, Funding acquisition, Project administration, Writing – review and editing

### Author ORCIDs

Chan Hee J Choi http://orcid.org/0000-0002-9892-9330
Mascha Koenen http://orcid.org/0000-0002-1024-4506
Zeran Lin http://orcid.org/0000-0003-4418-2443
Henrik Molina http://orcid.org/0000-0001-8950-4990
Paul Cohen http://orcid.org/0000-0002-2786-8585

### Ethics

All animal studies were performed in accordance with the institutional guidelines of the Rockefeller University Institutional Animal Care and Use Committee (IACUC) protocol (21010-H). Experiments involving adenoviral and AAV8 vectors were performed under general anesthesia using isoflurane, in accordance with the institutional ABSL-2 guidelines.

### Decision letter and Author response

Decision letter https://doi.org/10.7554/eLife.81559.sa1
Author response https://doi.org/10.7554/eLife.81559.sa2

## Additional files

### Supplementary files

- Supplementary file 1. Proteins detected in adipocyte CM with enrichment score above 80%.
- Supplementary file 2. Comparison of MS-based adipose secretome studies.
- Supplementary file 3. Primer sequences used for RT-qPCR analysis.

• MDAR checklist

## Data availability

The proteomic dataset has been deposited to Proteomexchange PRIDE under accession PXD035318. RNA-Seq data have been deposited to GEO under accession GSE208219. All original gels and blots are available as Source Data Files.

The following datasets were generated:

The following dataset was generated:

| Author(s) | Year | Dataset title | Dataset URL | Database and Identifier |
|---|---|---|---|---|
| Molina H | 2022 | LRG1 is an adipokine that promotes insulin sensitivity and suppresses inflammation | https://www.ebi.ac.uk/pride/archive/projects/PXD035318 | PRIDE, PXD035318 |
| Choi CHJ, Cohen P | 2022 | AAV-mediated LRG1 overexpression in diabetic mice | https://www.ncbi.nlm.nih.gov/geo/query/acc.cgi?acc=GSE208219 | NCBI Gene Expression Omnibus, GSE208219 |

The following previously published datasets were used:

| Author(s) | Year | Dataset title | Dataset URL | Database and Identifier |
|---|---|---|---|---|
| Shen Y, Ren B | 2012 | RNA-seq from ENCODE/LICR | https://www.ncbi.nlm.nih.gov/geo/query/acc.cgi?acc=GSE36026 | NCBI Gene Expression Omnibus, GSE36026 |
| Ai Su, Wiltshire T, Batalov S, Lapp H | 2004 | tissue-specific pattern of mRNA expression | https://www.ncbi.nlm.nih.gov/geo/query/acc.cgi?acc=GSE1133 | NCBI Gene Expression Omnibus, GSE1133 |

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

# Appendix 1

## Appendix 1—key resources table

| Reagent type (species) or resource | Designation | Source or reference | Identifiers | Additional information |
|---|---|---|---|---|
| Gene (*Mus musculus*) | *Lrg1* | NCBI | NM_029796.2 | |
| Strain, strain background (*Mus musculus*) | C57BL/6 J | The Jackson Laboratory | Cat #000664, RRID:IMSR_ JAX:000664 | |
| Strain, strain background (*Mus musculus*) | BKS.Cg-Dock7m +/+Leprdb/J | The Jackson Laboratory | Cat# 000642, RRID:IMSR_ JAX:000642 | |
| Strain, strain background (*Mus musculus*) | LRG1-KO in C57BL/6 J background | This Study | | Produced by the CRISPR and Genome Editing Center (Rockefeller University) |
| Strain, strain background (*Escherichia coli*) | NEB Stable | NEB | Cat# C3040 | Competent cells |
| Strain, strain background (*Escherichia coli*) | BJ5183-AD-1 cells | Agilent | Cat# 200157 | Electroporation competent, recombination proficient cells carrying the pAdEasy-1 plasmid |
| Strain, strain background (*Escherichia coli*) | XL-10 Gold ultracompetent cells | Agilent | Cat# 200314 | Competent cells |
| Strain, strain background (*Adenovirus*) | Ad-eGFP | This study; protocol adapted from *Luo et al., 2007* | | Produced in Paul Cohen lab (Rockefeller University) |
| Strain, strain background (*Adenovirus*) | Ad-LRG1-FL | This study; protocol adapted from *Luo et al., 2007* | | Produced in Paul Cohen lab (Rockefeller University) |
| Strain, strain background (*AAV8*) | AAV8-eGFP | This study | | Produced by Penn Vector Core (University of Pennsylvania). |
| Strain, strain background (*AAV8*) | AAV8-LRG1-FL | This study | | Produced by Penn Vector Core (University of Pennsylvania). |
| Cell line (*Homo-sapiens*) | HEK293A | Invitrogen | Cat# R70507 | |
| Antibody | Anti-LRG1 (rabbit polyclonal) | Sigma-Aldrich | Cat# HPA001888, RRID:AB_1079276 | Western blot (1:5000) |
| Antibody | Anti-CFD/Adipsin (sheep polyclonal) | R and D Systems | Cat# AF5430, RRID:AB_1655868 | Western blot (1:1000) |
| Antibody | Anti-Cytochrome *c* (rabbit monoclonal) | Cell Signaling Technology | Cat# 11940, RRID:AB_2637071 | Western blot (1:1000) |
| Antibody | Anti-FLAG M2 Magnetic Beads (mouse monoclonal) | Sigma-Aldrich | Cat# M8823, RRID:AB_2637089 | Immunoprecipitation |
| Antibody | Anti-Cleaved Caspase-3 (Asp175) (rabbit polyclonal) | Cell Signaling Technology | Cat# 9661, RRID:AB_2341188 | WB (1:1000) |
| Antibody | Anti-Caspase-3 (rabbit polyclonal) | Cell Signaling Technology | Cat# 9662, RRID:AB_331439 | WB (1:1000) |
| Antibody | Anti-IRS-1 (rabbit polyclonal) | Cell Signaling Technology | Cat# 2382, RRID:AB_330333 | WB (1:500) |
| Antibody | Anti-β-actin (rabbit polyclonal) | GeneTex | Cat# GTX109639, RRID:AB_1949572 | WB (1:20000) |
| Antibody | Anti-vinculin (rabbit polyclonal) | Cell Signaling Technology | Cat# 4650, RRID:AB_10559207 | WB (1:1000) |
| Peptide, recombinant protein | Recombinant Mouse M-CSF | Biolegend | Cat# 576406 | |

*Appendix 1 Continued on next page*

*Appendix 1 Continued*

| Reagent type (species) or resource | Designation | Source or reference | Identifiers | Additional information |
|---|---|---|---|---|
| Peptide, recombinant protein | Recombinant human LRG1 | R and D Systems | Cat# 7890-LR | |
| Peptide, recombinant protein | Recombinant human TNFα | R and D Systems | Cat# 210-TA-020/CF | |
| Peptide, recombinant protein | Cytochrome *c* from equine heart | Sigma-Aldrich | Cat# C2506 | |
| Commercial assay or kit | Click-iT Protein Enrichment Kit | Invitrogen | Cat# C10416 | |
| Sequence-based reagent | qPCR primers | This paper | | *Supplementary file 3* |
| Sequence-based reagent | pAdTrack-CMV | Addgene | RRID:Addgene_16405 | |
| Sequence-based reagent | pENN.AAV.CB7.CI.eGFP.WPRE.rBG | Addgene | RRID:Addgene_105542 | |

