## [Editor Report]

This paper presents a fundamental advance by elucidating the function of LRG1 as an adipokine. The authors provide compelling evidence for the biological effects of LRG1 on metabolism and its potential connections to metabolic diseases.

---

## [Decision Letter]

**Decision letter after peer review:**

Thank you for submitting your article "LRG1 is an adipokine that promotes insulin sensitivity and suppresses inflammation" for consideration by *eLife*. Your article has been reviewed by 2 peer reviewers, including Peter Tontonoz as the Reviewing Editor and Reviewer #1, and the evaluation has been overseen by David James as the Senior Editor.

The reviewers have discussed their reviews with one another, and the Reviewing Editor has drafted this to help you prepare a revised submission. The reviewers and editors expect that the comments raised can be addressed with additional discussion and limited new data.

Essential revisions:

1) It would be helpful to provide characterization of the physiology of LRG1 overexpressing mice under normal diet if this data is available.

2) Please quantify all blots. (especially Cytc blots)

3) The authors claim that LRG1 improves insulin sensitivity during obesity. However, it would be helpful to provide better evidence to support this, such as showing activation of insulin signaling pathways in liver and fat (Figure 4,5)

4) Please show released Cytc level peaks at week 7 in db/db mice and week 12 in HFD mice, as they suggest sharp increase of adipocyte death at this time point.

5) To support the idea that enhanced clearance of Cytc in Lrg1 KO mice blunts the outcome of HFD please show Cytc levels in the serum of Lrg1 KO mice during HFD.

6) Please add M2 macrophage markers along with M1 macrophage to evaluate whether LRG1 is controlling macrophage polarization (Figure 7).

7) Please address comments in public review with additional discussion as appropriate.

*Reviewer #1:*

The manuscript by Choi et al. describes the identification of novel adipokine LRG1, via non-canonical amino acid labeling in vivo. The authors show that LRG1 is an obesity-regulated adipokine and that its expression improves glucose tolerance in obese mice. These effects are mechanistically linked to the ability of LRG1 to dampen the inflammatory effect of cyt-c. Overall these studies rigorously characterize a circulating metabolic and inflammatory modulator with potential therapeutic relevance.

The data is convincing and clearly presented and the manuscript is well polished. This paper will be of broad interest to the field of molecular metabolism.

*Reviewer #2:*

This study reveals LRG1 as a novel adipokine and suggested a protective role in obesity-induced inflammation by binding to extracellular Cytochrome c. They discovered LRG1 using BONCAT and mass spectrometry, which minimizes distractions produced by different conditions. Overexpression of LRG1 in both HFD-induced obesity and db/db mice revealed that LRG1 improves glucose handling, but does not affect adipogenesis. Interestingly, the data demonstrated that overexpression of LRG1 suppresses macrophage infiltration into adipose tissue in both mouse models, along with decreased cytokine/ chemokine levels. It has previously been shown that LRG1 binds to Cytc, leading them to focus on this interaction to elucidate the molecular mechanism of LRG1 action. They showed that dead adipocytes release Cytc during obesity, which activates macrophages through TLR4 receptors. They suggested that increased LRG1 captures extracellular Cytc and decreases Cytc's potential to activate macrophages, maintaining insulin sensitivity by alleviating adipose tissue inflammation.

These data add another layer of complexity to the role of adipokines in obesity. Their precise approach to analyze adipokines revealed LRG1 and expanded the range of the adipocyte secretome. By overexpressing LRG1 in two different obesity model (HFD and db/db), they showed potential role of LRG1 in glucose homeostasis.

Issues:

1) The data indicates that overexpressing LRG1 in adipocytes can suppress obesity-induced adipose tissue inflammation. However, they also showed that LRG1 is increased in obesity (Figure 3). They should clarify this discrepancy as to why increased LRG1 in obesity is unable to inhibit adipose tissue inflammation.

2) The authors claim that LRG1 improves insulin sensitivity during obesity. However, they should provide better evidence to support this idea, such as showing activation of insulin signaling pathways in liver and fat (Figure 4,5).

3) The authors suggest that overexpressing LRG1 in db/db mouse improves glucose handling and suppresses adipose tissue inflammation (Figure 5). However, the protective effect disappears in 10 week old mice with increased body weight, even though inflammation in adipose tissue is still suppressed. The physiology of LRG1 overexpression in db/db mice probably requires more characterization.

4) The authors claim that overexpressed LRG1 decreases extracellular Cytc level. However, the data supporting their idea is not clear. They should provide more convincing data (Fig7.I). In addition, they should rule out the possibility that LRG1 decreases extracellular Cytc level by directly suppressing cell death to support their idea (Figure 7.H).

5) The data show that LRG1 decreases the opportunity of Cytc to activate macrophages through TLR4 signaling (Figure 7). However, how macrophages infiltrate or undergo activation are different process. Adding whether Cytc directly or indirectly (through inducing chemokines from adipocytes) recruits macrophage and LRG1 works on this step would support authors' conclusion more clearly.

---

## [Author Response]

Essential revisions:1) It would be helpful to provide characterization of the physiology of LRG1 overexpressing mice under normal diet if this data is available.

We agree that characterizing the effect of LRG1 overexpression in lean mice would provide a more comprehensive picture of its metabolic effects. Because LRG1 is regulated by obesity, we prioritized studying its function in mice on HFD. We currently do not have data describing LRG1 overexpression in mice on a chow diet. Such experiments will take 6-12 months to complete, and we feel this is beyond the scope of the current manuscript. In our future work, we plan to continue to characterize LRG1 function in various physiologic and disease settings.

2) Please quantify all blots. (especially Cytc blots)

We have now quantified blots that contain replicates and performed necessary statistical tests to support our arguments. We did not quantify some blots where qualitative assessment was deemed sufficient for data interpretation due to (1) lack of replicates or (2) presence of undetected bands.

3) The authors claim that LRG1 improves insulin sensitivity during obesity. However, it would be helpful to provide better evidence to support this, such as showing activation of insulin signaling pathways in liver and fat (Figure 4,5)

To further demonstrate that LRG1 overexpression improves glycemic profiles by insulin sensitization, we have added (1) HOMA-IR calculations and (2) IRS^-1^ western blots from eWAT and liver of *db/db* mice to the revised manuscript.

HOMA-IR is a widely used method for assessing insulin resistance (Matthews et al., 1985). In LRG1-overexpressing mice, we observed a trend towards lower HOMA-IR in the B6 DIO cohort (*P* = 0.059, Figure 4—figure supplement 1I) and a significantly lower HOMA-IR in the *db/db* cohort (*P* < 0.001, Figure 5—figure supplement 1G). These results provide further evidence that LRG1 improves insulin sensitivity.

To directly assess whether LRG1 affects the insulin signaling pathway, we performed western blots of phospho-AKT (pAKT) and total IRS^-1^ in *db/db* eWAT and liver. Our assessment of pAKT was limited by the fact that the mice were not stimulated with insulin prior to sacrifice. Basal pAKT levels after 6 hours of fasting showed no difference in either eWAT or liver (see Author response image 1). IRS^-1^ protein levels have been shown to be decreased in humans with diabetes (Rondinone et al., 1997) and in *ob/ob* mice (Kerouz et al., 1997). We observed that LRG1 overexpression leads to increased IRS^-1^ protein levels in *db/db* mice, especially in eWAT (Figure 5—figure supplements 1J,K). Given the critical role IRS^-1^ plays in insulin signaling, these results provide direct evidence that LRG1’s effect on glycemic profiles involves the insulin signaling pathway.

**Author response image 1. sa2fig1:** 

4) Please show released Cytc level peaks at week 7 in db/db mice and week 12 in HFD mice, as they suggest sharp increase of adipocyte death at this time point.

We have shown in Figure 7B (now quantified in Figure 7—figure supplement 1A) that Cyt *c* levels are increased in 7-week-old *db/db* mice compared to *misty* controls. We observed more variable but overall lower levels of serum Cyt *c* at 10 weeks of age.

We have also now measured circulating Cyt *c* levels in B6 mice for an extended duration of HFD feeding. During 5, 10, and 15 weeks on HFD, circulating Cyt *c* levels continued to diverge between chow and HFD mice (Figure 7—figure supplements 1C,D).

5) To support the idea that enhanced clearance of Cytc in Lrg1 KO mice blunts the outcome of HFD please show Cytc levels in the serum of Lrg1 KO mice during HFD.

We measured endogenous Cyt *c* levels in the serum of LRG1-KO and their WT littermates on 20 weeks of HFD and observed that Cyt *c* is virtually undetectable in the KO serum (Figure 7—figure supplement 1N). Together with the data showing enhanced clearance of injected Cyt *c* in the KOs, this result supports our hypothesis that renal excretion of Cyt *c* in LRG1-KO mice could blunt their metabolic phenotype.

6) Please add M2 macrophage markers along with M1 macrophage to evaluate whether LRG1 is controlling macrophage polarization (Figure 7).

We performed qPCR analysis on markers of M2 polarization and observed no consistent effect of either Cyt *c* or LRG1 treatment (Figure 7—figure supplement 1H).

7) Please address comments in public review with additional discussion as appropriate.

Please see below for our responses to the comments. Where indicated, we have also updated the main text of the manuscript to provide additional explanations and clarifications.

Reviewer #1:The manuscript by Choi et al. describes the identification of novel adipokine LRG1, via non-canonical amino acid labeling in vivo. The authors show that LRG1 is an obesity-regulated adipokine and that its expression improves glucose tolerance in obese mice. These effects are mechanistically linked to the ability of LRG1 to dampen the inflammatory effect of cyt-c. Overall these studies rigorously characterize a circulating metabolic and inflammatory modulator with potential therapeutic relevance.The data is convincing and clearly presented and the manuscript is well polished. This paper will be of broad interest to the field of molecular metabolism.

We thank the reviewer for recognizing our work as “convincing and clearly presented.” We appreciate your comments and believe the revised manuscript has further clarified our findings on LRG1.

Reviewer #2:This study reveals LRG1 as a novel adipokine and suggested a protective role in obesity-induced inflammation by binding to extracellular Cytochrome c. They discovered LRG1 using BONCAT and mass spectrometry, which minimizes distractions produced by different conditions. Overexpression of LRG1 in both HFD-induced obesity and db/db mice revealed that LRG1 improves glucose handling, but does not affect adipogenesis. Interestingly, the data demonstrated that overexpression of LRG1 suppresses macrophage infiltration into adipose tissue in both mouse models, along with decreased cytokine/ chemokine levels. It has previously been shown that LRG1 binds to Cytc, leading them to focus on this interaction to elucidate the molecular mechanism of LRG1 action. They showed that dead adipocytes release Cytc during obesity, which activates macrophages through TLR4 receptors. They suggested that increased LRG1 captures extracellular Cytc and decreases Cytc's potential to activate macrophages, maintaining insulin sensitivity by alleviating adipose tissue inflammation.These data add another layer of complexity to the role of adipokines in obesity. Their precise approach to analyze adipokines revealed LRG1 and expanded the range of the adipocyte secretome. By overexpressing LRG1 in two different obesity model (HFD and db/db), they showed potential role of LRG1 in glucose homeostasis.

Thank you very much for your thorough review of our work. We have added new data and explanations to address all of the concerns raised. We hope you agree that doing so has greatly strengthened our manuscript.

Issues:1) The data indicates that overexpressing LRG1 in adipocytes can suppress obesity-induced adipose tissue inflammation. However, they also showed that LRG1 is increased in obesity (Figure 3). They should clarify this discrepancy as to why increased LRG1 in obesity is unable to inhibit adipose tissue inflammation.

Indeed, our findings show that while LRG1 is induced in obesity, further increasing LRG1 expression via AAV transduction is protective. These findings suggest that LRG1 induction is physiologic, but the degree of increase observed in obesity is insufficient to confer full protection. In this context, AAV-mediated overexpression could be augmenting the body’s physiologic response by raising LRG1 concentration both in the local tissue microenvironment and in the circulation. We have clarified this point in the discussion (line 871).

2) The authors claim that LRG1 improves insulin sensitivity during obesity. However, they should provide better evidence to support this idea, such as showing activation of insulin signaling pathways in liver and fat (Figure 4,5).

Thank you for this suggestion. We have now addressed this in Point 3 of the essential revisions.

3) The authors suggest that overexpressing LRG1 in db/db mouse improves glucose handling and suppresses adipose tissue inflammation (Figure 5). However, the protective effect disappears in 10 week old mice with increased body weight, even though inflammation in adipose tissue is still suppressed. The physiology of LRG1 overexpression in db/db mice probably requires more characterization.

As glycemic profiles start to converge around week 10, differences in inflammatory gene expression profiles (Figure 6G) and circulating cytokine levels (Figure 6—figure supplement 1H) also become less dramatic. The persistence of histologic findings at week 10 therefore suggests a delay in histological changes compared to other signs of inflammation and glucose handling.

We agree that the constellation of metabolic effects caused by LRG1 overexpression needs further investigation and this will be the focus of our future work.

4) The authors claim that overexpressed LRG1 decreases extracellular Cytc level. However, the data supporting their idea is not clear. They should provide more convincing data (Fig7.I). In addition, they should rule out the possibility that LRG1 decreases extracellular Cytc level by directly suppressing cell death to support their idea (Figure 7.H).

Our intention was to further explore the finding in Figure 7A that LRG1-overexpressing mice have reduction in circulating Cyt *c*. Because our data suggest that LRG1 inhibits renal Cyt *c* clearance, reduction in serum Cyt *c* in the setting of LRG1 overexpression must be due to decreased release from tissues. Measurement of Cyt *c* in the interstitial fluid (Figure 7I) points towards metabolic tissues such as eWAT and liver as potential sources of this reduction. Together with the data in Figure 7H that provides direct evidence for reduction of cell death in the LRG1 group, we felt comfortable arguing that LRG1 overexpression decreases extracellular Cyt *c* as part of its immunomodulatory role.

We agree that mechanistically, various possible molecular phenomena could underlie our observed phenotype with LRG1 overexpression. We have not specifically assessed whether LRG1 can directly suppress cell death, but we also have not noticed changes in cell viability when treating primary adipocytes or BMDMs with LRG1-depleted FBS or recombinant LRG1. As we continue to work on the in vivo significance LRG1-Cyt *c* interaction as well as LRG1 function independent of Cyt *c*, further elucidation of the molecular mechanisms will also be an object of our future work.

We have added further explanation of these points to the main text (lines 915-926).

5) The data show that LRG1 decreases the opportunity of Cytc to activate macrophages through TLR4 signaling (Figure 7). However, how macrophages infiltrate or undergo activation are different process. Adding whether Cytc directly or indirectly (through inducing chemokines from adipocytes) recruits macrophage and LRG1 works on this step would support authors' conclusion more clearly.

While our current data with cultured BMDMs provide some information about Cyt *c*’s ability to activate macrophages, we agree that Cyt *c* may act on macrophages at various other stages of immune response. We will continue to explore how LRG1 and Cyt *c* affect macrophage recruitment in our future work. We have explicitly stated the limitations of our findings in the discussion (lines 920-926).

References

1. Kerouz, N.J., Hörsch, D., Pons, S., Kahn, C.R., 1997. Differential regulation of insulin receptor substrates^-1^ and -2 (IRS^-1^ and IRS-2) and phosphatidylinositol 3-kinase isoforms in liver and muscle of the obese diabetic (ob/ob) mouse. J Clin Invest 100, 3164–3172. https://doi.org/10.1172/JCI119872

2. Matthews, D.R., Hosker, J.P., Rudenski, A.S., Naylor, B.A., Treacher, D.F., Turner, R.C., 1985. Homeostasis model assessment: insulin resistance and β-cell function from fasting plasma glucose and insulin concentrations in man. Diabetologia 28, 412–419. https://doi.org/10.1007/BF00280883

3. Rondinone, C.M., Wang, L.-M., Lonnroth, P., Wesslau, C., Pierce, J.H., Smith, U., 1997. Insulin receptor substrate (IRS) 1 is reduced and IRS-2 is the main docking protein for phosphatidylinositol 3-kinase in adipocytes from subjects with non-insulin-dependent diabetes mellitus. Proceedings of the National Academy of Sciences 94, 4171–4175. https://doi.org/10.1073/pnas.94.8.4171